# Parallel SnowModel (v1.0): a parallel implementation of a Distributed Snow-Evolution Modeling System (SnowModel)

Ross Mower[1,2], Ethan D. Gutmann[1], Glen E. Liston[3], Jessica Lundquist[2], Soren Rasmussen[1]

[1]The NSF National Center for Atmospheric Research, Boulder, Colorado, USA

[2]Department of Civil and Environmental Engineering, University of Washington, Seattle, Washington, USA

[3]Cooperative Institute for Research in the Atmosphere, Colorado State University, Fort Collins, Colorado, USA

*Correspondence to*: Ross Mower (rossamower@ucar.edu)

**Abstract.** SnowModel, a spatially distributed, snow-evolution modeling system, was parallelized using Coarray Fortran for high-performance computing architectures to allow high-resolution (1 m to 100s of meters) simulations over large, regional to continental scale, domains. In the parallel algorithm, the model domain was split into smaller rectangular sub-domains that are distributed over multiple processor cores using one-dimensional decomposition. All the memory allocations from the original code were reduced to the size of the local sub-domains, allowing each core to perform fewer computations and requiring less memory for each process. Most of the subroutines in SnowModel were simple to parallelize; however, there were certain physical processes, including blowing snow redistribution and components within the solar radiation and wind models, that required non-trivial parallelization using halo-exchange patterns. To validate the parallel algorithm and assess parallel scaling characteristics, high-resolution (100 m grid) simulations were performed over several western United States domains and over the contiguous United States (CONUS) for a year. The CONUS scaling experiment had approximately 70% parallel efficiency; runtime decreased by a factor of 1.9 running on 1800 cores relative to 648 cores (the minimum number of cores that could be used to run such a large domain because of memory and time limitations). CONUS 100 m simulations were performed for 21 years (2000 – 2021) using 46,238 and 28,260 grid cells in the *x* and *y* dimensions, respectively. Each year was simulated using 1800 cores and took approximately 5 hours to run.

## 1 Introduction

The cryosphere (snow and ice) is an essential component of Arctic, mountain, and downstream ecosystems, Earth's surface energy balance, and freshwater resource storage (Huss et al., 2017). Globally, half the world's population depends on snowmelt (Beniston, 2003). In snow-dominated regions like the Western United States, snowmelt contributes to approximately 70% of the total annual water supply (Foster et al., 2011). In these regions, late-season streamflow is dependent on the deepest snow drifts and therefore longest-lasting snow (Pflug and Lundquist, 2020). Since modeling snow-fed streamflow accurately is largely dependent on our ability to predict snow quantities and the associated spatial and temporal variability (Clark and Hay, 2004), high-temporal and -spatial resolution snow datasets are important for predicting flood hazards and managing freshwater resources (Immerzeel et al., 2020).

The spatial and temporal seasonal snow characteristics also have significant implications outside of water resources.
Changes in fractional snow-covered area affect albedo and thus atmospheric dynamics (Liston, 2004; Liston and Hall, 1995).
Avalanches pose safety hazards to both transportation and recreational activities in mountainous terrain; the prediction of
which requires high-resolution (meters) snow datasets (Morin et al., 2020; Richter et al., 2021). Additionally, the timing and
duration of snow-covered landscapes strongly influence how species adapt, migrate, and survive (Boelman et al., 2019;
Liston et al., 2016; Mahoney et al., 2018).
To date, the primary modes for estimating snow properties and storage have come from observation networks, satellite-based
sensors, and physically derived snow algorithms in land surface models (LSMs). However, despite the importance of
regional, continental, and global snow, estimates of snow properties over these scales remain uncertain, especially in alpine
regions where wind, snow, and topography interact (Boelman et al., 2019; Dozier et al., 2016; Mudryk et al., 2015).
Observation datasets used for spatial interpolation of snow properties and forcing datasets used in LSMs are often too sparse
in mountainous terrain to accurately resolve snow spatial heterogeneities (Dozier et al., 2016; Renwick, 2014). Additionally,
remotely sensed products have shown deficiencies in measuring snowfall rate (Skofronick-Jackson et al., 2013), snow-water
equivalent (SWE), and snow depth (Nolin, 2010), especially in mountainous terrain where conditions of deep snow, wet
snow, and/or dense vegetation may be present (Lettenmaier et al., 2015; Takala et al., 2011; Vuyovich et al., 2014).
However, LSMs using high-resolution inputs, including forcing datasets from regional climate models (RCMs), have
demonstrated realistic spatial distributions of snow properties (Wrzesien et al., 2018).
Many physical snow models have been developed either in stand-alone algorithms or larger LSMs with varying degrees of
complexity based on their application. The more advanced algorithms attempt to accurately model snow properties at high
resolution especially in regions where snow interacts with topography, vegetation, and/or wind. Wind-induced snow
transport is one such complexity of snow that represents an important interaction between the cryosphere and atmosphere. It
occurs in regions permanently or temporarily covered by snow, influences snow properties (e.g. heterogeneity, sublimation,
avalanches, and melt timing), and has been shown to improve simulated snowpack distribution (Bernhardt et al., 2012;
Freudiger et al., 2017; Keenan et al., 2023; Quéno et al., 2023). Models that have incorporated wind-induced physics
generally require components to both develop the snow mass balance and incorporate atmospheric inputs of the wind field.
Additionally, these models typically require high resolution grids (1 to 100 m) as the redistribution components of the model
become negligible at larger spatial discretizations (Liston et al., 2007). However, there often exists a trade-off between the
accuracy of simulating wind-induced snow transport and the computational requirements for downscaling and developing
the wind fields over the gridded domain (Reynolds et al., 2021; Vionnet et al., 2014). Therefore, simplifying assumptions of
uniform wind direction has been applied in models like Distributed Blowing Snow Model (DBSM) (Essery et al., 1999; Fang
and Pomeroy, 2009). More advanced models have utilized advection-diffusion equations, like Alpine3D (Lehning et al.,
2006) or spatial distributed formulations like SnowTran-3D (Liston and Sturm, 1998). Finite volume methods for more
efficiently discretizing wind fields have been applied to models such as DBSM (Marsh et al., 2020). The most complex
models consider nonsteady turbulence which utilize three-dimensional wind fields from atmospheric models to simulate
blowing snow transport and sublimation; for example, SURFEX in Meso-NH/Crocus (Vionnet et al., 2014; Vionnet et al.,
2017), wind fields from the atmospheric model ARPS (Xue et al., 2000) being incorporated into Alpine3D (Mott and
Lehning, 2010; Mott et al., 2010; Lehning et al., 2008), and SnowDrift3D (Prokop and Schneiderbauer, 2011). Incorporating
wind-induced physics into snow models is computationally expensive; thus, parallelizing the serial algorithms would likely
be beneficial to many models.
For several decades, a distributed snow-evolution modeling system (SnowModel) has been developed, enhanced, and tested
to accurately simulate snow properties across a wide range of landscapes, climates, and conditions (Liston and Elder, 2006a;
Liston et al., 2020). To date, SnowModel has been used in over 200 refereed journal publications; a short listing of these is
provided by Liston et al. (2020). Physically derived snow algorithms, as used in SnowModel, that model the energy balance,
multilayer snow physics, and lateral snow transport are computationally expensive. In these models, the required
computational power increases with the number of grid cells covering the simulation domain. Finer grid resolutions usually
imply more grid cells and higher accuracy resulting from improved representation of process physics at higher resolutions.
The original serial SnowModel code was written in Fortran 77 and could not be executed in parallel using multiple processor
cores. As a result, SnowModel's spatial and temporal simulation domains (number of grid cells and time steps) were
previously limited by the speed of one core and the memory available on the single computer. Note that a "processor" refers
to a single central processing unit (CPU) and typically consists of multiple cores, each core can run one or more processes in
parallel.
Recent advancements in multiprocessor computer technologies and architectures have allowed for increased performance in
simulating complex natural systems at high resolutions. Parallel computing has been used on many LSMs to reduce compute
time and allow for higher accuracy results from finer grid simulations (Hamman et al., 2018; Miller et al., 2014). Our goal
was to develop a parallel version of SnowModel (Parallel SnowModel) using Coarray Fortran (CAF) syntax without making
significant changes to the original SnowModel code physics or structure. CAF is a Partitioned Global Address Space
(PGAS) programming model and has been used to run atmospheric models on 100,000 cores (Rouson et al., 2017).
In parallelizing numerical models, a common strategy is to decompose the domain into smaller sub-domains that get
distributed across multiple processes (Dennis, 2007; Hamman et al., 2018). For rectangular gridded domains (like
SnowModel), this preserves the original structure of the spatial loops and utilizes direct referencing of neighboring grids
(Perezhogin et al., 2021). The parallelization of many LSMs involve "embarrassingly parallel" problems requiring minimal
to no processor communication (Parhami, 1995); in this case, adjacent grid cells do not communicate with each other (an
example of this would be where each grid cell represents a point, or one-dimension, snowpack model that is not influenced
by nearby grid cells).
While much of the SnowModel's logic can be considered "embarrassingly parallel", SnowModel also contains "non-trivial"
algorithms within the solar radiation, wind, and snow redistribution models. Calculations within these algorithms often
require information from neighboring grid cells, either for spatial derivative calculations or for horizontal fluxes of mass
(e.g., saltating or turbulent-suspended snow) across the domain. Therefore, non-trivial parallelization requires implementing

99 algorithm changes that allow computer processes to communicate and exchange data. The novelty of the work presented

100 here includes 1) the presentation of Parallel SnowModel, high-resolution (100 m) distributed snow datasets over CONUS,

101 and an analysis of the performance of the parallel algorithm; 2) demonstrating how a simplified parallelization approach

102 using CAF and one-dimensional decomposition can be implemented in geoscientific algorithms to scale over large domains;

103 and 3) demonstrating an approach for non-trivial parallelization algorithms that involve spatial derivatives and fluxes using

104 halo-exchange techniques.

105 In Sect. 2, we provide background information on SnowModel, parallelization using CAF, data and domains used in this

106 study, and a motivation for this work. In Sect. 3, we explain our parallelization approach using CAF and introduce the

107 simulation experiments used to demonstrate the performance of Parallel SnowModel through strong scaling metrics and

108 CONUS simulations. In Sect. 4, we provide results of the simulation experiments introduced in Sect. 3. Lastly, we end with a

109 discussion in Sect. 5 and a conclusion in Sect. 6.

## 110 2 Background

### 111 2.1 SnowModel

112 SnowModel is a spatially distributed snow-evolution modeling system designed to model snow states (e.g., snow depth,

113 SWE, snow melt, snow density) and fluxes over different landscapes and climates (Liston and Elder, 2006a). The most

114 complete and up-to-date description of SnowModel can be found in the Appendices of Liston et al. (2020). While many

115 snow modelling systems exist, SnowModel will benefit from parallelization because of its ability to simulate snow processes

116 on a high-resolution grid through downscaling meteorological inputs and modelling snow redistribution. SnowModel is

117 designed to simulate domains on a structured grid with spatial resolutions ranging from 1 to 200 m (although it can simulate

118 coarser resolutions, as well) and temporal resolutions ranging from 10 m to 1 d. The primary modeled processes include

119 accumulation from frozen precipitation; blowing-snow redistribution and sublimation; interception, unloading, and

120 sublimation within forest canopies; snow-density and grain-size evolution; and snowpack ripening and melt. These processes

121 are distributed into four, core interacting submodules: MicroMet defines the meteorological forcing conditions (Liston and

122 Elder, 2006b), EnBal describes surface and energy exchanges (Liston, 1995; Liston et al., 1999), SnowPack-ML is a

123 multilayer snowpack sub-model that simulates the evolution of snow properties and the moisture and energy transfers

124 between layers (Liston and Hall, 1995; Liston and Mernild, 2012), and SnowTran-3D calculates snow redistribution by wind

125 (Liston et al., 2007). Additional simulation features include SnowDunes (Liston et al., 2018) and SnowAssim (Liston and

126 Hiemstra, 2008), which model sea-ice applications and data assimilation techniques, respectively. Figure 1 shows a

127 schematic of the core SnowModel toolkit. Additionally, the initialization submodules that read in the model parameters,

128 distribute inputs across the modeled grid, allocate arrays, etc., include PreProcess and ReadParam. Outputting arrays is

129 contained within the Outputs submodule. SnowModel incorporates first-order physics required to simulate snow evolution

130    within each of the global snow classes [e.g., Ice, Tundra, Boreal Forest, Montane Forest, Prairie, Maritime, and Ephemeral;
131    (Sturm and Liston, 2021; Liston and Sturm, 2021)].

132

**Figure 1: The original figure from Pedersen et al. (2015) was modified for the present paper, providing an example of possible inputs, core submodules, and outputs of SnowModel.**

## 2.2 Coarray Fortran

CAF, formerly known as F-, (Iso/Iec, 2010; Numrich and Reid, 1998; Numrich et al., 1997) is the parallel language feature of Fortran that was used to parallelize SnowModel. CAF is like Message Passing Interface (MPI) libraries in that it uses the Single Program Multiple Data (SPMD) model where multiple independent cores simultaneously execute a program. SPMD allows for distributed memory allocation and remote memory transfer. However, unlike MPI, CAF uses the PGAS parallel programming model to handle the distribution of computational tasks amongst processes (Coarfa et al., 2005). In the PGAS model, each process contains local memory that can be accessed directly by all other processes. While CAF and MPI syntax often refers to processes as images or ranks, for consistency, we will continue to use the term "process". Ultimately, CAF offers a high-level syntax that exploits locality and scales effectively (Coarfa et al., 2005). For simulation comparisons, we used OpenCoarrays, a library implementation of CAF (Fanfarillo et al., 2014) utilized by the gfortran compiler; intel and cray compilers both have independent CAF implementations.

## 2.3 Model Domains, Data, and Computing Resources

The required inputs for SnowModel include 1) temporally varying meteorological variables of precipitation, wind speed and direction, air temperature, and relative humidity taken from meteorological stations or atmospheric models and 2) spatially distributed topography and land-cover type (Liston & Elder, 2006a). The following inputs were used for the experiments introduced in Sect. 3: USGS National Elevation Dataset (NED) for topography (Gesch et al., 2018), The North American Land Change Monitoring System (NALCMS) Land Cover 2015 map for vegetation (Homer et al., 2015; Jin et al., 2019; Latifovic et al., 2016), and forcing variables from either the North American Land Data Assimilation System (NLDAS-2) (Mitchell, 2004; Xia, 2012a, b) on a 1/8 degree (approximately 12 km) grid or a high-resolution Weather Research Forecast

(WRF) model from the National Center for Atmospheric Research (NCAR) on approximately a 4 km grid (Rasmussen et al.,
2023). The high-performance computing architectures used include NCAR's Cheyenne supercomputer, which is a 5.43-
petaflop SGI ICE XA Cluster featuring 145,152 Intel Xeon processes in 4,032 dual-socket nodes and 313 TB of total
memory (Laboratory, 2019) and The National Aeronautics and Space Administration's (NASA) Center for Climate
Simulation (NCCS) Discover supercomputer with a 1,560-teraflop SuperMicro Cluster featuring 20,800 Intel Xeon Skylake
processes in 520 dual-socket nodes and 99.84 TB of total memory. Simulation experiments were conducted over six domains
(Tuolumne, CO Headwaters, Idaho, PNW, Western US, and CONUS) throughout the United States at 100 m grid resolution.
The spatial location, domain dimensions (e.g., number of grids in the *x* and *y* dimensions), and memory requirements,
derived from the peak_memusage package (https://github.com/NCAR/peak_memusage), for the simulation experiments are
highlighted in Fig. 2.

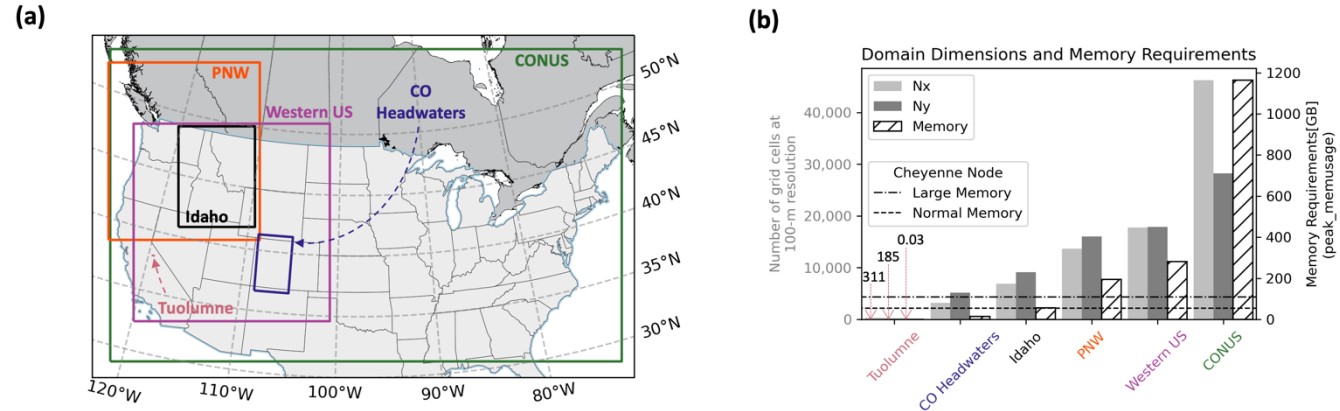


**Figure 2: (a)** *Spatial location of simulated domains on WRF's lambert conformal projection (Rasmussen et al., 2023) and (b)*
*corresponding grid dimensions (Nx – number of grids in x dimension; Ny – number of grids in y dimension) and memory obtained*
*from peak_memusage package required for single-layer SnowModel simulation experiments. For reference, the dashed lines represent*
*the normal and large memory thresholds (55 and 109 GB) for Cheyenne's SGI ICE XA cluster.*
**2.4 Parallelization Motivation**
The answers to current snow science, remote sensing, and water management questions require high-resolution data that
covers large spatial and temporal domains. While modeling systems like SnowModel can be used to help provide these
datasets, running them on single-processor workstations imposes limits on the spatiotemporal extents of the produced
information. Serial simulations are limited by both execution time and memory requirements, where the memory limitation
is largely dependent on the size of the simulation domain. Up to the equivalent of 175 two-dimensional and 10 three-
dimensional arrays are held in memory during a SnowModel simulation, depending on the model configuration. In analyzing
the performance of the Parallel SnowModel (Sect. 4), serial simulations were attempted over six domains throughout the
United States at 100 m grid resolution (Fig. 2) for the 2018 water year (1 September 2017 to 1 September 2018). Only the
Tuolumne domain could be simulated in serial based on the memory (109 GB for a large memory node) and time (12 h wall-
clock limit) constraints on Cheyenne. The CO Headwaters and Idaho domains could not be simulated in serial due to time
constraints, while the three largest domains (Pacific Northwest (PNW), Western U.S. and CONUS) could not be executed in
serial due to both exceedances of the 12 h wall-clock limit and memory availability. Furthermore, we estimate that using a
currently available, state of the art, single-processor workstation, would require approximately 120 d of computer time to
perform a 1 y model simulation over the CONUS domain. SnowModel is regularly used to perform multi-decade
simulations, for trend analyses, climate change studies, and retrospective analyses (Liston and Hiemstra, 2011; Liston et al.,
2020; Liston et al., 2022). If this 1 y, 100 m, CONUS domain was simulated for a 40 y period (e.g., 1980 through present), it
would take approximately 4800 d, or over 13 y, of computer time. Clearly such simulations are not practical using single-
processor computer hardware and software algorithms.
**3 Methods**
In parallelizing SnowModel and distributing computations and memory over multiple processes, we demonstrate its ability
to efficiently run regional to continental sized simulations. Some of the model configurations were not parallelized for
reasons including ongoing development in the serial code base and limitations to the parallelization approach. These
configurations are further discussed in Appendix A. This section introduces the syntax and framework used to parallelize
SnowModel and the simulation experiments used to assess the performance of the parallel algorithm.
**3.1 Parallel Implementation**
Changes to the SnowModel logic were made through the parallelization process and included the partitioning algorithm,
non-trivial communication via halo-exchange, and file input and output (I/O) schemes.
**3.1.1 Partitioning Algorithm**
The partitioning strategy identifies how the workload gets distributed amongst processes in a parallel algorithm. The
multidimensional arrays of SnowModel are stored in row-major order, meaning the $x$ dimension is contiguous in memory.
Additionally, dominant wind directions and therefore predominant snow redistribution occurs in the east-west direction as
opposed to south-north directions. Therefore, both the data structures and physical processes involved in SnowModel justify
a one-dimensional decomposition strategy in the $y$ dimension, where the computational global domain $N_x$ x $N_y$ is separated
into $N_x$ x $l_{ny}$ blocks. If $N_y$ is evenly divisible by the total number of processes (N), $l_{ny} = N_y / N$. If integer division is
not possible, the remaining rows are distributed evenly amongst the processes starting at the bottom of the computational
domain. Figure 3 demonstrates how a serial domain containing 10 grid cells in the $x$ and $y$ dimensions would be
decomposed with four processes using our partitioning strategy.

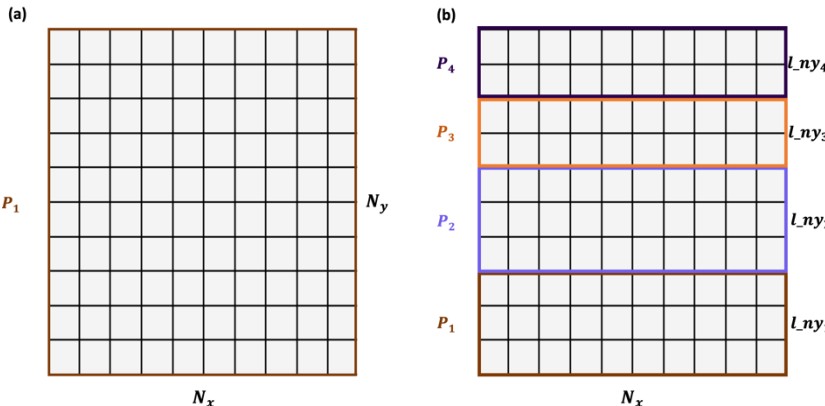


**Figure 3:** *Example 10 x 10 global domain and partitioning for (a) serial simulation and (b) parallel simulation using four processes.*

### 3.1.2 Non-trivial Parallelization

Each process has sufficient information to correctly execute most of the physical computations within SnowModel. However, there are certain subroutines where grid computations require information from neighboring grid cells (e.g., data dependencies) and therefore information outside of the local domain of a process. For SnowModel, these subroutines typically involve the transfer of blowing snow or calculations requiring spatial derivatives. Furthermore, with our one-dimensional decomposition approach, each grid cell within a process local domain has sufficient information from its neighboring grid cells in the $x$ dimension but potentially lacks information from neighboring grid cells in the $y$ dimension. As a regular grid method, SnowModel lends itself to process communication via halo-exchange where coarrays are used in remote calls. Halo-exchange using CAF involves copying boundary data into coarrays on neighboring processes and using information from the coarrays to complete computations (Fig. 4). Although the entire local array could be declared a coarray and accessed by remote processes more directly, some CAF implementations, (e.g. Cray) impose additional constraints upon coarray memory allocations that can be problematic for such large allocations.

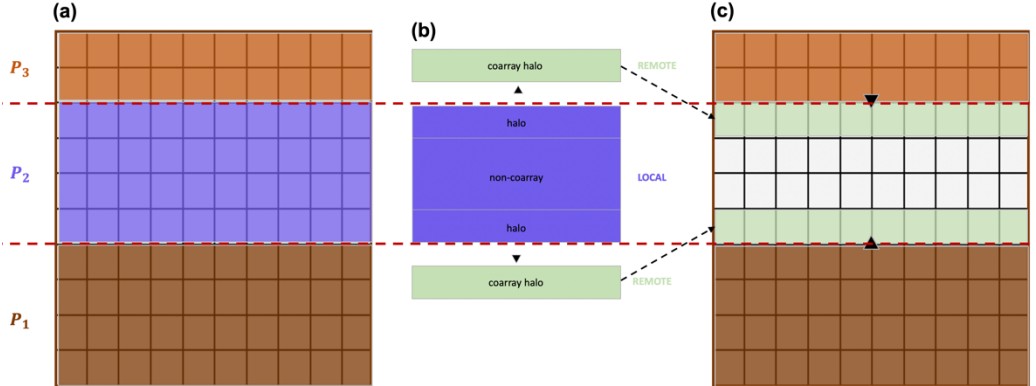


**Figure 4:** *Schematic showing halo-exchange using coarrays. The steps include: (a) initial gridded representation of local arrays for*
*three processes, (b) $P_2$ copying boundary data into coarrays for remote access, (c) neighboring processes ($P_1$ and $P_3$) stitching coarray*
*to local domains.*

### 3.1.2.1 Topography – Wind and Solar Radiation Models

The wind and solar radiation models in MicroMet require information about surrounding surface topography (Liston and Elder, 2006b). The wind model requires surface curvature, and the solar radiation model requires surface slope and aspect. These vary at each timestep as snow accumulates and melts because the defined surface includes the snow surface on top of the landscape. The surface curvature, for example, is computed at each model grid cell using the spatial gradient of the topographic elevation of eight neighboring grid cells. Using the parallelization approach discussed above, processes lack sufficient information to make curvature calculations for the bordering grid cells along the top and/or bottom row(s) within their local domains. Note that the number of row(s) (`inc`) is determined by a predefined parameter that represents the wavelength of topographic features within a domain. Future work should permit this parameter to vary spatially to account for changes in the length scale across the domain. For example, all grid cells along the top row of $P_1$ will be missing information from nearby grid cells to the north and require topographic elevation (`topo`) information from the bottom row(s) of the local domain of $P_2$ to make the calculation (Fig. 5a). Halo-exchange is performed to distribute row(s) of data to each process that is missing that information in their local domains (Fig. 5b). Processes whose local domains are positioned in the bottom or top of the global domain will only perform one halo-exchange with their interior neighbor, while interior processes will perform two halo-exchanges. By combining and appropriately indexing information from the process local array and received coarrays of topographic elevation, an accurate curvature calculation can be performed using this parallel approach (Fig. 5c).

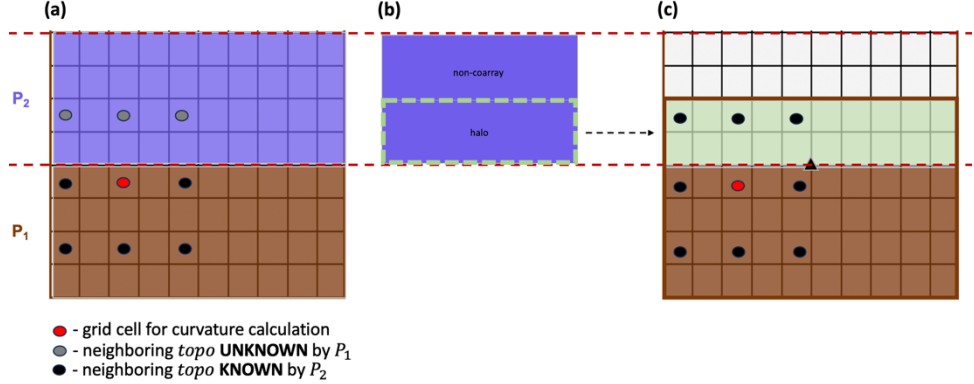

● - grid cell for curvature calculation
◯ - neighboring *topo* **UNKNOWN** by $P_1$
● - neighboring *topo* **KNOWN** by $P_2$

**Figure 5: Schematic for halo-exchange used in the curvature calculation by $P_1$, where `inc` = 2. (a) Prior to halo-exchange, $P_1$ contains insufficient information to perform the curvature calculation, (b) grid cells (halo) within the local domain of $P_2$ are (c) transferred to $P_1$ via coarrays. At this point, $P_1$ has sufficient information to make the curvature calculation.**

### 3.1.2.2 Snow Redistribution

Wind influences the mass balance of the snowpack by suspending and transporting snow particles in the air (turbulent-suspension) and by causing snow grains to bounce on top of the snow surface (saltation). In SnowModel, the saltation and suspension algorithms are separated into northerly, southerly, easterly, and westerly fluxes based on the $u$ and $v$ components of wind direction for each grid cell. Figure 6 shows a simplified schematic for the saltation flux from a southerly wind. In the

serial algorithm (Fig. 6a), SnowModel initializes the saltation flux based on the wind speed at that time step (`initial`
`flux`). To calculate the final saltation flux (`updated flux`), SnowModel steps through regions of continuous wind
direction (delineated by the indices: `jstart` and `jend`), updates the change in saltation fluxes from upwind grid cells and
the change in saltation flux from the given wind direction, and makes adjustments to these fluxes based on the soft snow
availability above the vegetation height (Liston and Elder, 2006a). Similar logic is used for the parallel implementation of
the saltation and suspension fluxes with an additional iteration (*salt iter*) that updates the boundary condition for each
process via halo-exchange. This allows the fluxes to be communicated from the local domain of one process to another. To
minimize the number of iterations, *salt iter* was provided a maximum bound that is equivalent to snow being
transported 15 km via saltation or suspension. This number was chosen based off prior field measurements (Tabler, 1975)
and simulation experiments. It is possible that in other environments an even larger length may be required. To be
guaranteed to match the serial results in all cases, the number of iterations would have to be equal to the number of
processes; however, this would result in no parallel speed up and has no practical benefit. A schematic of the parallel
calculation of the change in saltation due to southerly winds is illustrated in Fig. 6b. The *bc_halo_exchange* represents
a halo-exchange of grid cells from upwind processes, allowing the saltation flux to be transported from one process local
domain to the next.

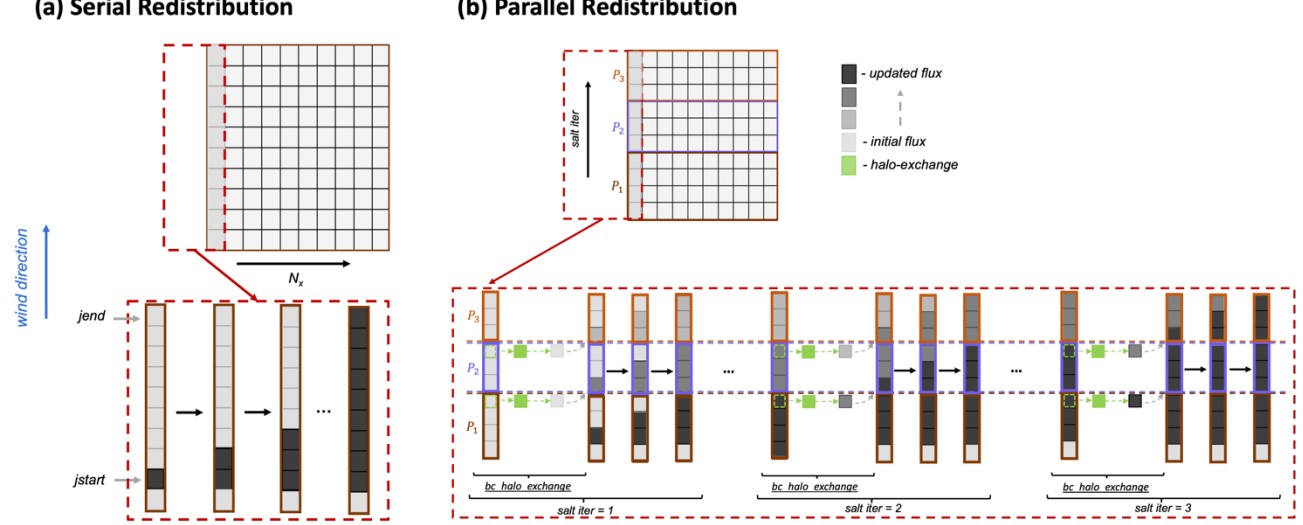


**Figure 6: (a)** *Schematic of the serial and **(b)** parallel redistribution algorithm showing the change in saltation flux due to southerly*
*winds over a gridded domain for $N_x$ = 1. The parallel schematic demonstrates how three processes ($P_1$, $P_2$, $P_3$) use an additional*
*iteration (`salt iter`) to perform a halo-exchange (`bc_halo_exchange`) and update the boundary condition of the saltation flux.*
**3.1.3 File I/O**
File I/O management can be a significant bottleneck in parallel applications. Parallel implementations that are less memory
restricted commonly use local to global mapping strategies, or a ***Centralized*** approach for file I/O (Fig. 7a). This approach
requires that one or more processes stores global arrays for input variables and that one process (Process 1; Fig. 7a) stores

global arrays for all output variables. As the domain size increases, the mapping of local variables to global variables for outputting creates a substantial bottleneck. To improve performance, ***Distributed*** file I/O can be implemented, where input and output files are directly and concurrently accessed by each process (Fig. 7b).

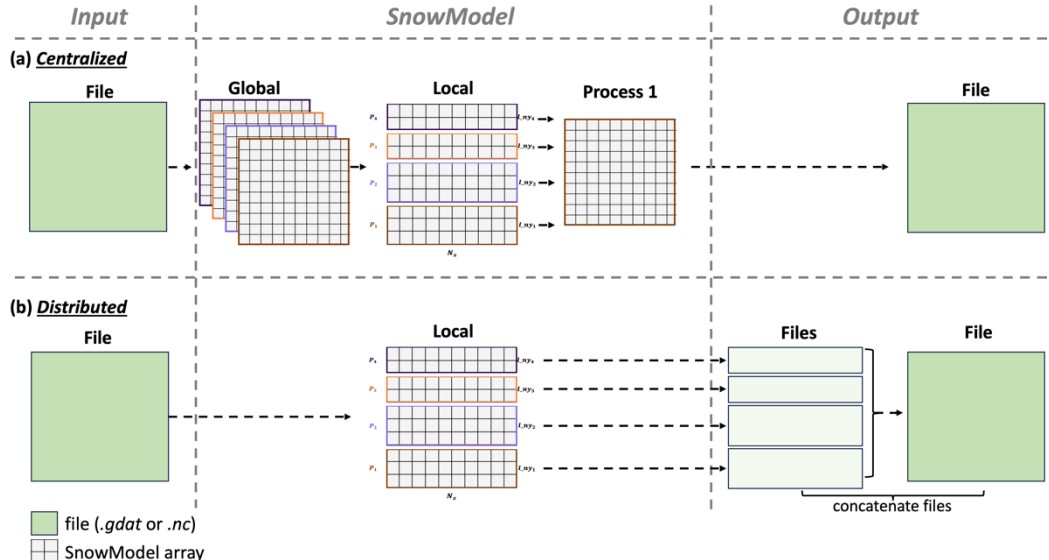

**Figure 7: (a)** *Schematic of global to local mapping for file I/O using a Centralized approach with four processes, and (b) Distributed file I/O where each process reads and writes data corresponding to its local domain.*

SnowModel contains static spatial inputs that do not vary over time (e.g., topography and land cover) and dynamic spatial inputs (e.g., air temperature and precipitation) that vary spatially and temporally. The static inputs are of a higher resolution compared to the dynamic inputs (cf., topography is on the model grid, while atmospheric forcing is almost always more widely spaced). To balance performance and consistency with the serial logic of the code, we used a mixed parallel file I/O approach. A goal of this work was to maintain nearly identical serial and parallel versions of the code in one code base that can be easily maintained and utilized by previous, current, and future SnowModel users with different computational resources and skills. Therefore, we wanted to maintain both the *Centralized* and *Distributed* file I/O approaches. However, for optimal parallel performance over larger simulation domains, file input (reading) is performed in a *Distributed* way for the static inputs and in a *Centralized* way for dynamic inputs, while file output (writing) is performed in a *Distributed* way, as described further below. This permits the new version of the code to be a drop in replacement for the original serial code without requiring users to install new software libraries or manage hundreds of output files, while enabling users who wish to take advantage of the parallel nature of the code to do so with minimal additional work and no changes to the underlying code.

**3.1.3.1 Parallel Inputs**

As noted above, SnowModel has two primary types of input files, temporally static files such as vegetation and topography and transient inputs such as meteorological forcing data. While acceptable static input file types include flat binary, NetCDF,

and ASCII files for the serial version of the code, optimizing the efficiency of Parallel SnowModel requires static inputs from binary files that can be accessed concurrently and directly subset by indexing the starting byte and length of bytes commensurate to a process local domain. Therefore, each process can read its own portion of the static input data. For very large domains, the available memory becomes a limitation when using the centralized approach. For example, the CONUS simulation could not be simulated using a centralized file I/O approach because each process would be holding global arrays of topography and vegetation in memory, each of which would require approximately 5.2 GB of memory per process.

Reading of meteorological forcing variables (wind speed, wind direction, relative humidity, temperature, and precipitation) can be performed in parallel with either binary or NetCDF files. Depending on the forcing dataset, the grid spacing of the meteorological variables typically ranges from 1 to 30 km and therefore often requires a smaller memory footprint than static inputs for high-resolution simulations. For example, the resolution of NLDAS-2 meteorological forcing has a grid of approximately 11 km, while the high-resolution WRF model used has a 4 km grid. At each timestep, processes read in the forcing data from every station within the domain into a one-dimensional array, index the nearest locations for each SnowModel grid, and interpolate the data to create forcing variables over the local domain. All processes perform the same operation and store common information; however, since the resolutions of the forcing datasets are significantly coarser than the model grid for high-resolution simulations, the dynamic forcing input array size remains comparable to other local arrays and does not impose significant memory limitations for simulations performed to date. While more efficient parallel file input schemes could improve performance, we decided to keep this logic in part to maintain consistency with the serial version of the code and minimize code changes.

**3.1.3.2 Parallel Outputs**

To eliminate the use of local to global mapping commonly used to output variables (Fig. 7a), each process writes its own output file (Fig. 7b). A postprocessing script is then used to concatenate files from each process into one file that represents the output for the global domain. Modern high-performance computing architectures have highly parallelized storage systems making file output using a distributed approach significantly faster than the centralized approach. Therefore, file output in this manner reduces time and memory requirements. Future work could leverage other established parallel I/O libraries at the cost of additional installation requirements.

**3.2 Simulation Experiments**

Parallel SnowModel experiments were conducted to both evaluate the effectiveness of the parallelization approach used in this study (Sect. 3.1) and to produce a high-resolution snow dataset over CONUS. All experiments were executed with a 100 m grid increment, a 3 h time step, a single-layer snowpack configuration, and included the primary SnowModel modules (MicroMet, EnBal, SnowPack, and SnowTran-3D). These experiments are further described below, with results provided in Sect. 4.

Validation experiments comparing output from the original serial version of the code to the parallel version were conducted continuously throughout the parallel algorithm development to assess the reproducibility of the results. Additionally, a more

thorough validation effort was performed at the end of the study that compared output from the serial algorithm to that of the
parallel algorithm, while varying the domain size, the number of processes, and therefore the domain decomposition. Results
from all of these validation experiments produced root mean squared error (RMSE) values of $10^{-6}$, which is at the limit of
machine precision, when compared to serial simulation results. See Appendix B for more details on the validation
experiments. The serial version of SnowModel has been evaluated in many studies across different snow classes (Sturm and
Liston, 2021; Liston and Sturm, 2021), time periods, and snow properties. Evaluations ranged from snow cover (Pedersen et
al., 2016; Randin et al., 2015), snow depth (Szczypta et al., 2013; Wagner et al., 2023), SWE (Freudiger et al., 2017;
Hammond et al., 2023; Mortezapour et al., 2020; Voordendag et al., 2021), and SWE-melt (Hoppinen et al., 2023; Lund et
al., 2022), using field observations, snow-telemetry stations, and remote sensing products. A full comparison of the Parallel
SnowModel simulations presented here with observations across CONUS is beyond the scope of the present work.
Incorrectly simulated SWE could affect the scaling results and CONUS visualizations presented in Sect. 3.2.1.1, 3.2.1.2, and
3.2.2; for example, if zero SWE were incorrectly simulated in many locations, processing time would be less than if SWE
had been simulated and tracked. However, based on the scale of these analyses and the fact that SnowModel has been
previously evaluated in a wide range of locations, we believe the impacts of this limitation on the computational results
presented here are minimal.

### 3.2.1 Parallel Performance

In high performance computing, scalability attempts to assess the effectiveness of running a parallel algorithm with an
increasing number of processes. Thus, scalability can be used to identify the optimal number of processes for a fixed domain,
understand the limitations of a parallel algorithm as a function of domain size and number of processes, and estimate the
efficiency of the parallel algorithm on new domains or computing architectures. Speedup, efficiency, and code profiling
were tools used to assess the scalability and performance of Parallel SnowModel on fixed domains. Speedup [`S(N)`; Eq.
(1)], a metric of strong scaling, is defined as the ratio of the serial execution time, `T(1)`, over the execution time using `N`
processes, `T(N)`. Optimally, parallel algorithms will experience a doubling of speedup as the number of processes is
doubled. Some reasons why parallel algorithms do not follow ideal scaling include the degree of concurrency possible and
overhead costs due to communication. Synchronization statements have an associated cost of decreasing the speed and
efficiency of an algorithm due to communication overhead and requirements for one process to sit idle while waiting for
another to reach the synchronization point. Furthermore, speedup tends to peak or plateau at a certain limit on a given
computing architecture and domain because either the overheads grow with an increasing number of processes, or the
number of processes exceeds the degree of concurrency inherent in the algorithm (Kumar and Gupta, 1991). For large
domains, where serial simulations cannot be performed either due to wall-clock or memory limitations, relative speedup,
[$\hat{S}$`(N)`; Eq. (2)], is commonly used. Relative speedup is estimated as a ratio of the execution time, `T(`$\hat{P}$`)`, of the minimum
number of processes, (`$\hat{P}$`), that can be simulated on a given domain over `T(N)`. An additional speedup metric, approximate

speedup [Ṧ(N); Eq. (3)], is introduced to estimate S by assuming perfect scaling from $\widehat{P}$ to a single process. While this is only an approximation, it is helpful to compare the Ṧ across the different domains on a similar scale. Additionally, efficiency [E(N); Eq. (4)], and approximate efficiency [Ë(N); Eq. (5)] are the ratios of S to N and Ṧ to N, respectively. A simulation that demonstrates ideal scaling, would have 100% efficiency. Additionally, code profiling evaluates the cumulative execution time of individual submodules (e.g. Preprocess, Readparam, MicroMet, Enbal, SnowPack, SnowTran-3D, and Output) as a function of the number of processes. Together, code profiling and strong scaling can be used to understand locations of bottlenecks in the algorithm and how changes to the code enhance performance.

$$S(N) = \frac{T(1)}{T(N)} \qquad \text{Eq. 1}$$

$$\widehat{S}(N) = \frac{T(\widehat{P})}{T(N)} \qquad \text{Eq. 2}$$

$$\ddot{S}(N) = \frac{T(\widehat{P})}{T(N)} * \widehat{P} \qquad \text{Eq. 3}$$

$$E(N) = \frac{S}{N} * 100\% \qquad \text{Eq. 4}$$

$$\ddot{E}(N) = \frac{\ddot{S}}{N} * 100\% \qquad \text{Eq. 5}$$

### 3.2.1.1 Parallel Improvement

To better understand how changes to the Parallel SnowModel code have affected its performance, speedup and code profiling plots were assessed for simulations using three distinct versions of the code. These versions represent snapshots of the algorithms development and quantify the contributions of different types of code modifications to the final performance of the model. These versions were identified by different GitHub commits (Mower et al., 2023) and can be summarized as follows. The first or baseline version represents an early commit of Parallel SnowModel, where file I/O is performed in a *Centralized* way, as described in Sect. 3.1.3. Each process stores both a local and global array in memory for all input variables, makes updates to its local arrays, distributes that updated information into global arrays used by one process to write each output variable. The embarrassingly parallel portion of the physics code has been parallelized, but the snow redistribution step is not efficiently parallelized, it has a larger number of synchronizations and memory transfers. Therefore, this approach has significant time and memory constraints. The *Distributed* version represents an instance of the code where distributed file I/O (Sect. 3.1.3) had first been implemented. In this version, each process reads and writes input and output variables for its local domain only. Global arrays and the communication required to update these variables are no longer needed; this alleviates memory constraints and shows the value of parallelizing I/O in scientific applications. Lastly, the *Final* version represents the most recent version of Parallel SnowModel, (at the time of this publication) where the snow transport algorithm had been optimized to run efficiently. This was done by reducing unnecessary memory allocations,

reducing the transfer of data via coarrays, and optimizing memory transfers to reduce synchronization calls. This shows the
value of focused development on a single hotspot of the code base. The simulations were executed on the CO Headwaters
domain (Fig. 2) using 1, 2, 4, 16, 36, 52, 108, and 144 processes, outputted only a single variable, and were forced with
NLDAS-2 data from 23-24 March 2018. While 2-days is a short period to perform scaling experiments, a significant amount
of wind and frozen precipitation was observed over the CO Headwaters domain during the simulation to activate some of the
snow redistribution schemes in SnowTran-3D. Furthermore, to avoid disproportionately weighing the initialization of the
algorithm, we removed the timing values from the ReadParam and Preprocess submodules from the total execution time
used in the speedup analysis. Results from these experiments are provided in Sect. 4.1.
**3.2.1.2 Strong Scaling**
Strong scaling experiments of Parallel SnowModel were evaluated by comparing the approximate speedup and efficiency ($\ddot{S}$
and $\ddot{E}$) over six different size domains across the United States, all with a 100 m grid spacing [Tuolumne, CO Headwaters,
Idaho, PNW, Western U.S., and CONUS] (Fig. 2). These experiments use the *Final* version of the code according to Sect.
3.2.1.1. The simulations were forced with NLDAS-2 data for 2928 timesteps from 1 September 2017 to 1 September 2018
and output one variable (SWE). The number of processes used in these simulations varied by domain based on the 12 h wall-
clock and memory constraints on Cheyenne. Results from these experiments are provided in Sect. 4.2.
**3.2.2 CONUS Simulations**
A primary goal of this work was to run Parallel SnowModel simulations for 21 years (2000 – 2021) over the CONUS
domain (Fig. 2) on a 100 m grid, while resolving the diurnal cycle in the model physics and creating a daily dataset of snow
properties, including snow depth, SWE, melt rate, and sublimation. Future work will analyze results from these simulations.
The CONUS domain contained 46,238 and 28,260 grid cells in the *x* and *y* dimensions, respectively. Simulations were
performed on a 3 h time step and forced with the WRF dataset. All simulations were executed on Discover using 1800
processes with a total compute time of approximately 192,600 core hours, or approximately 5 wall-clock hours per year.
**4 Results**
**4.1 Parallel Improvement**
Figure 8 demonstrates how the scalability of Parallel SnowModel evolved, as shown through code profiling (top row; Fig. 8)
and speedup (bottom row; Fig. 8) plots at three different stages (*Centralized*, *Distributed*, and *Final*) of the code
development. The code profiling plots display the cumulative execution time of each submodule (`T(N)[log (s)]`) as a
function of the `N`. The strong scaling plots show the total execution time (`T(N)[s]`) and the speedup [`S(N)`; Eq. (1)] as a
function of `N` on the primary y-axis and secondary y-axis, respectively. As mentioned previously, the initialization timing
was removed from these values. The speedup of the *Centralized* version of the code quickly plateaus at approximately 10
processes. While the Enbal, SnowPack, and MicroMet subroutines scale with the number of processes (execution time
decreases proportional to the increase in the number of processes), the ReadParam, Preprocess, and Output subroutines,
which all perform file I/O or memory allocation, require a fixed execution time regardless of the number of processes used,
and the execution time of the SnowTran-3D submodule increases beyond 16 processes. This highlights the large bottleneck
that often occurs during the file I/O step in scientific code and the importance of code infrastructure outside of the physics
routines. In contrast, all the submodules in the *Distributed* version of the code, scale up to 36 processes, at which point the
inefficient parallelization of the SnowTran-3D submodule causes a significant slowdown, an increase in execution time as
the number of processes increases. This results in a speedup that plateaus at 52 processes and decreases beyond 108
processes. In the *Final* version of the code, scalability is observed well beyond 36 processes, with a maximum speedup of
100 observed using 144 processes. The execution time of all the submodules decreases as the number of processors
increases. This work highlights the value of going beyond the rudimentary parallelization of a scientific code base by
profiling and identifying individual elements that would benefit the most from additional optimization. This is a well-known
best practice in software engineering but often underappreciated in high-performance scientific computing. In Parallel
SnowModel, the improvement of these communication bottlenecks is primarily attributed to utilizing a distributed file I/O
scheme and minimizing processor communication by limiting the use of coarrays and synchronization calls. Ultimately,
without these improvements, the CONUS domain could not be simulated using Parallel SnowModel.

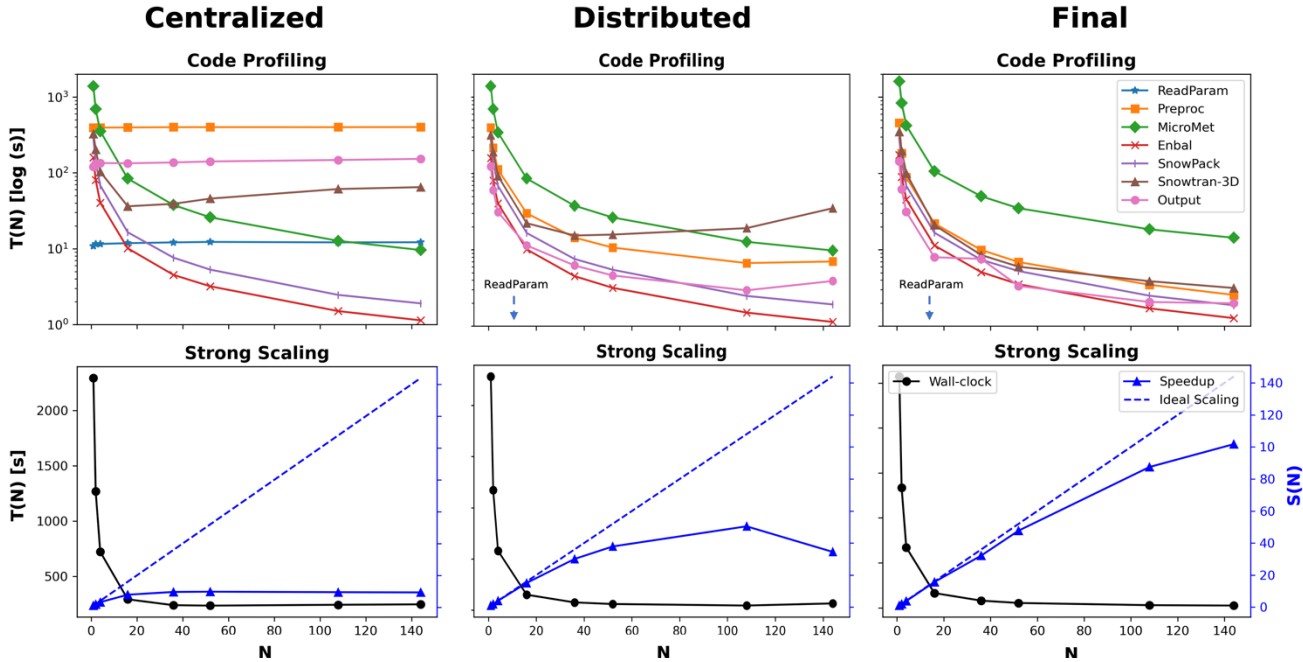


**Figure 8: Code profiling (top row) and strong scaling (bottom row) results demonstrating the progression of Parallel SnowModel,**
**which includes a version of the code with centralized file I/O (*Centralized*; first column), a version of the code with distributed file**
**I/O (*Distributed*; second column), and a final version of the code at the time of this publication (*Final*; third column). These**
**versions can be found as different commits within the GitHub repository (Mower et al., 2023). The code profiling plots display the**
**cumulative execution time of each submodule on a logarithmic scale as a function of the number of processes (N). The arrow in the**
**code profiling plots of *Distributed* and *Final* indicates the ReadParam timing is below the y-axis at approximately 0.3 seconds and**
**0.003 seconds, respectively. The strong scaling plots show the total execution time (`T(N)`) against `N` on the primary y-axis and the**
**speedup (`S`) against `N` on the secondary y-axis.**

## 4.2 Strong Scaling

In addition to the parallel improvement analysis, strong scaling was also performed on six domains for the 2018 water year
to better understand how Parallel SnowModel scales across different domain sizes and decompositions. Figure 9 displays the
approximate speedup [$\breve{S}$(`N`); Eq. (3)] of Parallel SnowModel for three local/state domains (Tuolumne, CO Headwaters, and
Idaho) and three regional/continental domains (PNW, Western US, and CONUS). Additionally, Table 1 contains information
about the minimum and maximum number of processors ($\widehat{P}$ and `P*`, respectively) simulated on each domain and their
corresponding execution time (`T(N)[m]`), relative speedup [$\widehat{S}$(`N`); Eq. (2)], approximate speedup [$\breve{S}$(`N`); Eq. (3)], and
approximate efficiency [$\ddot{E}$(`N`); Eq. (5)]. As mentioned previously, simulations were constrained by both the 12 h wall-clock
and 109 GB of memory per node on the Cheyenne supercomputer. In strong scaling, the number of processes is increased
while the problem size remains constant; therefore, it represents a reduced workload per process. Local-sized domains, e.g.,
Tuolumne, likely do not warrant the need for parallel resources because they have small serial runtimes (e.g., using 52
processes, Tuolumne had an $\ddot{E}$ of 38%; Table 1). However, state, regional, and continental domains stand to benefit more
significantly from parallelization. The CONUS runtime decreased by a factor of 3 running on 3456 processes relative to 648
processes. Based on our approximate speedup assumption, we would estimate a CONUS $\breve{S}$ of 1690 times on 3456 processes
compared to one process, with an $\ddot{E}$ of 49%. The Western US and PNW domains display very similar scalability results (Fig.
9), which is attributed to the similar number of grid cells in the `y` dimension (Fig. 2 and Table 1) and thus parallel
decomposition for each domain. Furthermore, these domains may also have a similar proportion of snow-covered grid cells.
While the PNW likely has more terrestrial grid cells that are covered by snow for a longer period throughout the water year,
it also has a significant number of ocean grid cells where snow redistribution would not be activated.

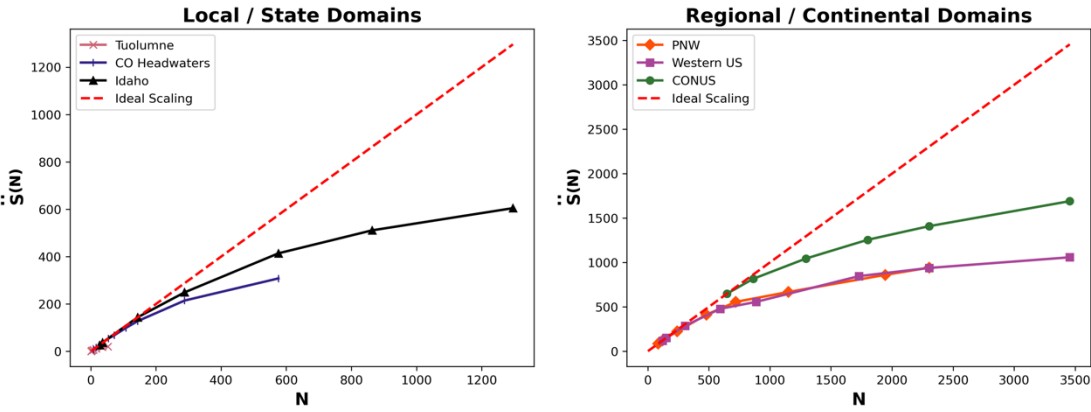


**Figure 9: The left panel displays approximate speedup as a function of the number of processes (N) for local and state sized simulations (Tuolumne, CO Headwaters, and Idaho), while the right panel shows $\ddot{S}$ for the regional and continental sized domains (PNW, Western US, and CONUS).**


| Domain | Nx | Ny | $\widehat{P}$ or $P\star$ | Number of Processes | Execution Time [m] | Relative Speedup | Approximate Speedup | Approximate Efficiency |
|---|---|---|---|---|---|---|---|---|
| | | | | N | $T(N)$ | $\widehat{S}(N)$ | $\ddot{S}(N)$ | $\ddot{E}(N)$ |
| Tuolumne | 311 | 185 | $\widehat{P}$ | 1 | 13 | --- | --- | 100 |
| | | | $P\star$ | 52 | 1 | 20 | 20 | 38 |
| CO Headwaters | 3166 | 5167 | $\widehat{P}$ | 8 | 934 | --- | 8 | 100 |
| | | | $P\star$ | 576 | 24 | 39 | 308 | 53 |
| Idaho | 6916 | 9107 | $\widehat{P}$ | 27 | 1068 | --- | 27 | 100 |
| | | | $P\star$ | 1296 | 48 | 22 | 605 | 47 |
| PNW | 13677 | 16058 | $\widehat{P}$ | 84 | 1173 | --- | 84 | 100 |
| | | | $P\star$ | 2304 | 105 | 11 | 941 | 41 |
| Western US | 17737 | 17878 | $\widehat{P}$ | 120 | 1187 | --- | 120 | 100 |
| | | | $P\star$ | 3456 | 135 | 9 | 1058 | 31 |
| CONUS | 46238 | 28260 | $\widehat{P}$ | 648 | 1196 | --- | 648 | 100 |
| | | | $P\star$ | 3456 | 459 | 3 | 1690 | 49 |


**Table 1: Parallel SnowModel strong scaling results containing grid dimensions (Nx and Ny), execution time [m], relative speedup, approximate speedup, and approximate efficiency for simulations executed with the minimum and maximum number of processes ($\widehat{P}$ and $P\star$, respectively) on the Tuolumne, CO Headwaters, Idaho, PNW, Western US, and CONUS domains. Values of the timing, speedup, and efficiency variables are rounded to the nearest integer.**

Strong scaling analysis is useful for I/O and memory bound applications to identify a setup that results in a reasonable runtime and moderate resource costs. Based on these scaling results, Fig. 10 contains the relationship between the number of processes (N) at which each domain is estimated to reach 50% $\ddot{E}$ (using linear interpolation) with the total number of grid

cells in the y dimension (Ny) and the average number of grid cells in the y dimension per process ($\mathtt{l_{ny}}$; inset Fig. 10). At this level of efficiency, it is notable the consistency of both the linear relationship between Ny and N (8.7:1 ratio) and the values of $\mathtt{l_{ny}}$ (5 to 11) for these year-long simulations that vary in both domain size and the proportion of snow-covered area. Similar relationships (Fig. 10) can be used to approximate the scalability of Parallel SnowModel on different sized domains and can be adjusted for the desired level of efficiency. For example, we decided to run the CONUS simulations (Sect. 4.3) using 1800 processes based on its 70% approximate efficiency.

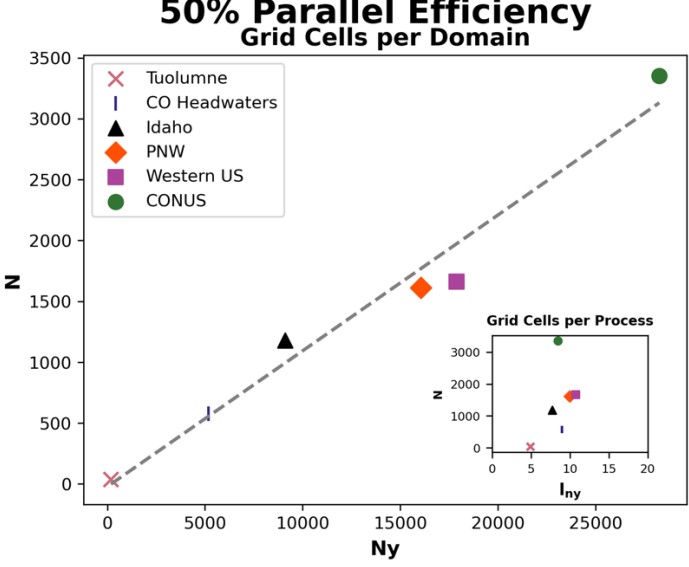

**Figure 10: Relationship between the number of grid cells in the y dimension (Ny) and the number of processes (N) for each domain at which 50% approximate efficiency is estimated using the strong scaling analysis. The dashed line represents the best fit line for this relationship using OLS regression. The inset figure displays a similar relationship but compares N to the average number of grid cells in the y dimension per process ($\mathtt{l_{ny}}$), instead of Ny.**

### 4.3 CONUS Simulations

Spatial results of SWE on 12 February 2011 over the CONUS domain and a sub-domain located in the Indian Peaks west of Boulder, Colorado are displayed in Fig. 11. On this date, simulated SWE was observed throughout the northern portion of the CONUS domain with the largest values concentrated in the mountain ranges (Fig. 11a). The Indian Peaks sub-domains of distributed SWE (Fig. 11b) with reference topography (Fig. 11c) underscores the ability of the large dataset to capture snow processes in a local alpine environment. It is important to note that while SnowModel does simulate snow redistribution, it does not currently have an avalanche model, which may be a limitation of accurately simulating SWE within this sub-domain. Additionally, Fig. 11b highlights two grid cells located 200 m apart on a peak. Figures 11d and 11e display the SWE evolution of these two grid cells over the entire dataset (water years 2000 – 2021) and the 2011 water year, respectively, further demonstrating the ability of Parallel SnowModel to capture fine-scale snow properties even when simulating continental domains. The upwind (western) grid cell is scoured by wind, and snow is transported to the downwind

(eastern) grid cells where a snow drift forms. The information and insight available in this high-resolution dataset will have
important implications for many applications from hydrology, to wildlife and ecosystems, to weather and climate, and many
more.

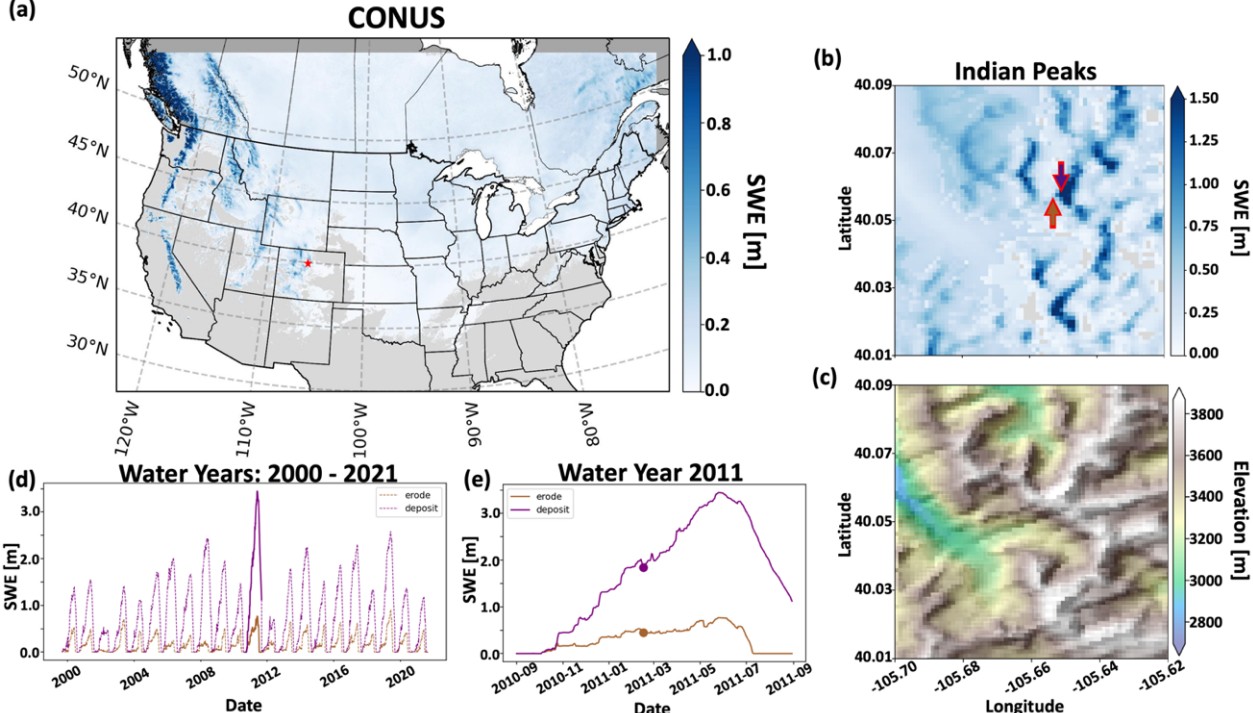

**Figure 11:** *Simulation results of Parallel SnowModel over CONUS using the WRF projection. (a) Spatial patterns of SWE over the CONUS domain for 12 February 2011, (b) highlighting the SWE distribution (c) and topography with an applied hillshade of a sub-domain near Apache Peak in the Indian Peaks west of Boulder, CO. (d) Time series of SWE from 2000-2021 and (e) over the 2011 water year for grid cells ("erode" and "deposit") identified in panel (b). The "erode" and "deposit" grid cells highlight areas of similar elevation but significant differences in SWE evolution resulting from blowing-snow redistribution processes.*

## 5 Discussion

Parallelizing numerical models often involves two-dimensional decomposition in both the *x* and *y* dimensions. While many
benefits have been demonstrated by this approach, including improved load balancing (Dennis, 2007; Hamman et al., 2018),
it comes with increased complication of the parallel algorithms, including the partitioning algorithm, file I/O, and process
communication. The demonstrated speedup (Fig. 9) suggests Parallel SnowModel scales effectively over regional to
continental domains using the one-dimensional decomposition approach. The added benefits obtained from two-dimensional
decomposition strategies might not outweigh the costs of development, testing, and minimizing changes to the code structure
and logic for applications such as SnowModel. Ultimately, our simplified parallelization approach can be implemented by
other geoscience schemes as a first step to enhance simulation size and resolution.

Simulation experiments were conducted using Parallel SnowModel to validate the parallel logic, interpret its performance across different algorithm versions and domain sizes, and demonstrate its ability to simulate continental domains at high-resolution. Code profiling and speedup analyses over the CO Headwaters domain helped identify bottlenecks in file I/O and processor communication in SnowTran-3D during the development of the parallel algorithm (Sect. 4.1). Corrections to the referred bottlenecks allowed Parallel SnowModel to scale up to regional and continental sized simulations and highlights the value of optimizing scientific code. For Parallel SnowModel scalability is primarily dependent on the number of grid cells per process ($Nx$ and $l_{ny}$) but is also affected by the proportion of snow-covered grid cells with sufficient winds and soft snow available to be redistributed (Sect. 3.1.2.2). The scalability analyses showed similar results across domains with significant differences in size ($Nx$ and $Ny$), topography, vegetation, and snow classifications (Sturm et al., 1995; Sturm and Liston, 2021) (Sect 4.2), highlighting the effectiveness of Parallel SnowModel for running state, regional, and continental-sized domains. Furthermore, results from this analysis can be used to estimate the number of processors required to simulate domains outside of the ones used in this study with a desired level of parallel efficiency (Fig. 10).

Additionally, these experiments emphasize the relationships among speed, memory, and computing resources for Parallel SnowModel. A common laptop (~ 4 processes) has sufficient CPUs to run local sized domains within a reasonable amount of time, but likely does not have sufficient memory for state-sized simulations. Similarly, the minimum memory (1160 GB; Fig. 1) required to run the CONUS domain, could be simulated on a large server (~ 128 processes) with one process per node. However, extrapolating from our scaling results on Cheyenne (Fig. 9), we estimate it would take over 2.5 days to run a CONUS simulation for one water year with this configuration. In contrast, it took approximately 5 hours for CONUS to run on the Discover supercomputer using 1800 processes. Therefore, by the time it took the large server to complete a CONUS simulation for one water year, 12 water years could have been simulated on a supercomputer. Lastly, results from the CONUS simulation highlight the ability of Parallel SnowModel to run high-resolution continental simulations, while maintaining fine-scale snow processes that occur at a local level (Sect. 4.3).

SnowModel can simulate high-resolution outputs of snow depth, density, SWE, grain size, thermal resistance, snow strength, snow albedo, landscape albedo, meltwater production, snow-water runoff, blowing snow flux, visibility, peak winter SWE, snow-season length, snow onset date, snow-free date, and more, all produced by a physical model that maintains consistency among variables. While several snow data products exist, few capture the suite of snow properties along with the spatio-temporal extents and resolutions that can benefit a wide variety of applications. For example, current snow information products include the NASA daily SWE distributions globally for dry (non-melting) snow on a 25 km grid (Tedesco and Jeyaratnam, 2019), a NASA snow-cover product on a 500 m grid (Hall et al., 2006) that is missing information due to clouds approximately 50% of the time (Moody et al., 2005), and the Snow Data Assimilation System (SNODAS) daily snow information provided by the National Oceanic and Atmospheric Administration (NOAA) and the National Weather Service (NWS) National Operational Hydrologic Remote Sensing Center (NOHRSC) on a 1 km grid (Center, 2004), which is itself model derived and has limited geographic coverage and snow properties. The Airborne Snow Observatory (ASO) provides the highest resolution data with direct measurements of snow depth on a 3 m grid, and derived values of SWE on a 50 m grid

(Painter et al., 2016), but has limited spatio-temporal coverage and a high cost of acquisition. Furthermore, there are many fields of study that can benefit from 100 m resolution information of internally consistent snow variables, including wildlife and ecosystem, military, hydrology, weather and climate, cryosphere, recreation, remote sensing, engineering and civil works, and industrial applications. The new Parallel SnowModel described here permits the application of this modeling system to very large domains without sacrificing spatial resolution.

## 6 Conclusions

In this paper, we present a relatively simple parallelization approach that allows SnowModel to perform high-resolution simulations over regional to continental sized domains. The code within the core submodules (EnBal, MicroMet, SnowPack, and SnowTran-3D) and model configurations (single-layer snowpack, multi-layer snowpack, binary input files, etc.) was parallelized and modularized in this study. This allows SnowModel to be compiled with a range of Fortran compilers, including modern compilers that support parallel CAF either internally or through libraries, such as OpenCoarrays (Fanfarillo et al., 2014). Additionally, it provides the structure for other parallelization logic (e.g., MPI) to be more easily added to the code base. The parallel module contains a simple approach to decomposing the computational domain in the $y$ dimension into smaller rectangular sub-domains. These sub-domains are distributed across processes to perform asynchronous calculations. The parallelization module also contains logic for communicating information among processes using halo-exchange coarrays for the wind and solar radiation models, as well as for snow redistribution. The scalability of Parallel SnowModel was demonstrated over different sized domains, and the new code enables the creation of high-resolution simulated snow datasets on continental scales. This parallelization approach can be adopted in other parallelization efforts where spatial derivatives are calculated or fluxes are transported across gridded domains.

## Appendix A

Some of the configuration combinations were not parallelized during this study for reasons including ongoing development in the serial code base and limitations to the parallelization approach. These include simulations involving tabler surfaces (Tabler, 1975), I/O using ASCII files, Lagrangian seaice tracking, and data assimilation.

## Appendix B

Validation SnowModel experiments were run in serial and in parallel over the Tuolumne and CO Headwaters domains (Sect. 4.1) using the RMSE statistic. Important output variables from EnBal, MicroMet, SnowPack, and SnowTran-3D demonstrated similar, if not identical values, when compared to serial results for all timesteps during the simulations; RMSE values were within machine precision ($\sim10^{-6}$) regardless of the output variable, domain, or number of processes used. The

validated output variables include albedo [%], precipitation [$m$], emitted longwave radiation [$W * m^{-2}$], incoming longwave radiation reaching the surface [$W * m^{-2}$], incoming solar radiation reaching the surface [$W * m^{-2}$], relative humidity [%], runoff from base of snowpack [$m * timestep$], rain precipitation [$m$], snow density [$kg * m^{-3}$], snow-water equivalent melt [$m$], snow depth [$m$], snow precipitation [$m$], static-surface sublimation [$m$], snow-water equivalent [$m$], air temperature [$°C$], wind direction [°], and wind speed [$m * s^{-1}$]. Ultimately, we feel confident that Parallel SnowModel is producing the same results as the original serial algorithm.

## Code, data availability, and supplement

The Parallel SnowModel code and the data used in Sect. 4 is available through a public GitHub repository (Mower et al., 2023). For more information about the serial version of SnowModel, refer to Liston and Elder (2006a). The data includes figures and SnowModel output files that contain the necessary information to recreate the simulations. The gridded output variables themselves are not included due to storage limitations. Pending approval, we will submit our code to get a DOI.

## Author contribution

EDG and GDL conceived the study. RM, EDG, GDL, and SR were integral in the code development. RM, EDG, and JL were involved in the design, execution, and interpretation of the experiments. All authors discussed the results and contributed to the final version of the draft.

## Competing interests

The contact author has declared that none of the authors has any competing interests.

## Disclaimer

Publisher's note: Copernicus Publications remains neutral with regard to jurisdictional claims in published maps and institutional affiliations.

## Financial support

This material is based upon work supported by the NSF National Center for Atmospheric Research, which is a major facility sponsored by the U.S. National Science Foundation under Cooperative Agreement No. 1852977. The authors would like to acknowledge that this work has been performed under funding from NASA Earth Science Office (ESTO) Advanced Information Systems Technology (AIST) Program (grant no. 80NSSC20K0207), support by the University of Washington's

College of Engineering Fellowship, and computational support from NSF  NCAR Computational and Information Systems
Lab (CISL) and NASA High-End Computing (HEC) Program through the NASA Center for Climate Simulation (NCCS) at
Goddard Space Flight Center.

**Acknowledgements**

We acknowledge Alessandro Fanfarillo in his help during the early stages of the Parallel SnowModel code development. We
are also grateful for the feedback from various team members involved in the AIST project, including Carrie Vuyovich,
Kristi Arsenault, Melissa Wrzesien, Adele Reinking, and Barton Forman.

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
