# Peer review of "Parallel SnowModel (v1.0): a parallel implementation of a Distributed Snow-Evolution Modeling System (SnowModel)"

_EGUsphere, 2023_

## Author Comment (AC1)

**Referee #1**

This is a review of "Parallel SnowModel (v1.0): a parallel implementation of a Distributed Snow-Evolution Modeling System (SnowModel)" where the authors describe imporovements made to Snowmodel by including a distributed parallel scheme. These improvements allow for running over large spatial extents and for longer temporal periods.

Broadly, I like what the authors have done to use CAF to bring parallelism to an existing code. Hydrology has long lacked HPC-aware models and the science has been poorer for it.

However, I struggle to follow aspects of this manuscript and I do not find the scaling results convincing. Finally, the lack of any validation against observations gives me pause, especially given the SWE results shown are almost certainly wrong for large portions of the domain in Figure 11. I will detail these concerns below.

We appreciate your support of this work and concern for some of the results. In terms of scaling, we will rerun the scaling results for a time period from 2017-09-01 to 2018-09-01 (2928 timesteps vs. the previous 16 timesteps) for more realistic timing comparisons. We will also extend our validation time period to incorporate both accumulation and ablation over the two domains.

In terms of the lack of validation, your comment made us dig into simulated SWE values from Figure 11 near Calgary in Alberta and identify a binning issue with how the data is being visualized. Therefore, most of Alberta should actually be represented as the 0.01 – 0.10m SWE bin. Thank you very much for this comment and apologies for the error. We will update the figure but acknowledge that it still might be error prone. However, this study focuses on parallelizing the algorithm, verifying that results are identical to simulations using the serial version of the code and demonstrating performance through strong scaling and CONUS simulations. We added an additional few sentences referencing how extensively the serial code has been validated. However, it is beyond the scope of this study to adequately validate Parallel SnowModel output on a CONUS scale.

First, the introduction essentially fails to cite any of the European or Canadian literature on snow dynamics, blowing snow, and the existing model developments that have been made. Notable contributions from Mott, Durand, Lehning, Vionnet, Marsh, Pomeroy, Musselman, MacDonald, Morin, Fang, and Essery to name but a few are all missing and would provide valuable context to the Liston, et al modelling efforts.

Thank you for addressing this oversight. We will add the following paragraph to the introduction.

Many physical snow models have been developed either in stand-alone algorithms or larger LSMs with varying degrees of complexity. The more advanced algorithms attempt to accurately simulate snow properties at higher resolution especially in regions where snow interacts with topography, vegetation, and/or wind. Wind-induced snow transport occurs in regions permanently or temporarily covered by snow and greatly influences snow heterogeneity, sublimation, avalanches, and melt timing. Models that have incorporated wind-induced physics generally require components to both develop the snow mass balance and incorporate atmospheric inputs of the wind field. However, there often exists a tradeoff between the accuracy of simulating wind-induced snow transport and the computational requirements for downscaling and developing the wind fields over the gridded domain (Reynold et al., 2021). Therefore, simplifying assumptions of uniform wind direction have been applied in models like a Distributed Blowing Snow Model (DBSM; Essery et al., 1999; Fang & Pomeroy, 2009). More advanced models have utilized advection-diffusion equations over spatially distributed uniform gridded wind fields, like SnowTran3D (Liston & Sturn, 1998) and unstructured grids using finite volume methods in DBSM (Marsh et al., 2020). The most complex models consider nonsteady turbulence which utilize three-dimensional wind fields from atmospheric models to simulate blowing snow transport and sublimation; for example, SURFEX in Meso-NH/Crocus (Vionnet et al., 2014, 2017), wind fields from the atmospheric model ARPS (Xue et al., 2000) being incorporated into Alpine3D (Mott and Lehning, 2010; Mott et al., 2010; Lehning et al., 2008), and SnowDrift3D (Schneiderbauer & Prokop, 2021).

Secondly, I find the mixing of methods and results to be very confusing. This is exceptionally bad in the Parallel Performance (S4.2) section where multiple code revisions are described. It is not at all clear where the different 'Distributed high Sync', etc are coming from. In some of these, the results presented are trivial — of course one would expect increased synchronization across more processes to incure scaling limitations. It is not clear if the SnowTran-3D plateau at 36-processes is the final code, or was a WIP code. I get the impression the authors are attempting to convey their profiling journey to optimize their code, but a) a general audience likely is not interested in all the specifics and b) it's confusing laid out leaving an interested reader muddled. For example on line 386, it is unclear /what/ versions of the code were even used. This section strikes me as the crux of the results, and is therefore an important section. However, I struggled to make my way through it. I would strongly sugges the authors split the methodology out and well describe what was profiled, etc and how this shaped the CAF implimentation. And then in the results, clearly and simply show "it is faster by XYZ for domains PQR".

Thank you for addressing this confusion, and we agree that the methods and results could be structured in a clearer way. We will restructure the methods to more briefly discuss the parallelization approach (e.g. 3.1 Parallel Implementation) and then introduce the different simulation experiments before discussing Results in Section 4. An updated structure for the methods and results is as follows:

3. Methods

    3.1.      Parallel Implementation

        3.1.1.      Partitioning Algorithm

        3.1.2.      Non-trivial Parallelization

            3.1.2.1.      Topography – Wind and Solar Radiation

            3.1.2.2.      Snow Redistribution

        3.1.3.    File I/O

            3.1.3.1.      Parallel Inputs

            3.1.3.2.      Parallel Outputs

    3.2.      Simulation Experiments

        3.2.1.      Parallel SnowModel Validation

        3.2.2.      Parallel SnowModel Performance

            3.2.2.1.      Strong Scaling

            3.2.2.2.      Code Profiling

            3.2.2.3.      CONUS Simulations

4. Results

    4.1.      Parallel SnowModel Validation

    4.2.      Parallel SnowModel Performance

        4.2.1.      Strong Scaling

        4.2.2.      Code Profiling

        4.2.3.      CONUS Simulations

After restructuring the methods and results, we will simplify the Code Progression or code profiling journey to see if the confusion can be alleviated. If not, we may add this to a supplemental section of the manuscript, instead. However, we do believe that these are important tools and results that can help others in their journey to parallelize legacy code.

In addition, the 16 timesteps are really not compelling as currently presented. I am sympathetic to the computational constraints. However, without code coverage, is there any guarantee that the code was tested in a representative manner? For example, if there were few or no melt / blowing snow (or if there was any snow!) then the results would not be typical of a run. This criticism exists for the 1 month serial v. distributed period (L333) as well. Is this a representative period of time viz a viz excercising the toughest numerical code paths (e.g., blowing snow and multilayer snowpacts, canopy interception) and highest sync code paths?

Thank you for this concern. We very much agree that this is not the best representation of scaling results. Therefore, we are going to run the six domains for 2928 timesteps (2017/09/01 - 2018/09/01) instead of the previous 16 timesteps. While this will affect the minimum number of processes that can be used, we believe this will be representative of sufficient synchronization opportunities across all domains.

My read is that Figure 10 is the "final" code that is evaluated for scaling testing. My following comments are through this lens. I do not find Figure 10 convincing of strong scaling. I would expect PNW to be the most difficult to simulate region with deep snow covers, and many blowing snow events. It performs weakly, with essentially plateaued scaling at 750 processes. As more non-blowing snow (and non-snow) cells are added in the CONUS domain, the scaling increases (shown in Figure 11). Essentially my read is the more non-snow cells that are added, the better the scaling. This is not a strong scaling result. Rephrased, over domains with significant snow processes, the scaling is poor.

Correct, Figure 10 represents the "final" code. We will make that more clear as we restructure the methodology for the simulation experiments. As mentioned, we are going to rerun the scaling results for 2928 timesteps (2017/09/01 - 2018/09/01) instead of the previous 16 timesteps. Thus, we will have to reevaluate the strong scaling results. Additionally, we will look at code profiling plots and more explicit communication timing profilers to see if we can identify how the domain decomposition affects communication requirements and thus scaling.

The simulated SWE results presented in Figure 11 are  suspect. This is total SWE on the ground in Feb, correct? In the middle of the winter (Feb) there is snow covering much of Canada — the foothills of Alberta, the Priaries of AB, SK, MB, and the Boreal forests of AB, SK, and MB. In the simulation results shown in Figure 11a, the domain east of continental

divid, including the eastern Rockies, is shown as having zero SWE. This is almost certainly not correct. The authors note that an evaluation of the SWE data will be done at a later point, but if this number of no-op grid cells are being used for the scaling evaluation, then the scaling evaluation is not representative of a real winter simulation.

As mentioned, we really appreciate this insight as it pointed us to a bug in our binning algorithm for visualizing the results. We will update the distributed plots of SWE figures (a) and (b) to reflect correct binning and a different color visualization to make the bins more easily distinguishable. As a result, we are observing SWE values between 0.01 and 0.10 m outside of Calgary and much of Alberta, for example. We also appreciate the implications of not simulating SWE correctly as it comes to scaling due to potential communication requirements across processes. Furthermore, we will add some discussion to this point as a limitation of our study. However, validating Parallel SnowModel for the simulated time period across CONUS is outside the scope of this study. We will focus on validating the serial and parallel versions of the code and highlight the many other studies that have validated the serial version code to insitu observations.

Figure 11e shows the erosion and then deposition across a ridgeline. However, in most mountain regions, this deposited snow will avalanche to a lower elevation. Given there is no avalanche model in this code and no avalanche literature is cited, these results are not compelling. Perhaps this is a ridgeline that doesn't have avalanches. But this needs to be noted if true.

Thank you for that observation. Correct, SnowModel does not contain an avalanche model. We will note that these time series and spatial results may be unrealistic of SWE because Parallel SnowModel lacks an avalanche model.

In conclusion, I like that the authors are describing making the code HPC-aware by using CAF with a simple halo exchange. I think there is value in showing the community that "legacy" models can be updated and that it is "not that hard." Such messaging has the potential to help normalize HPC-aware code development. However, the scaling results seem to show significant limitations in the scaling and the better CONUS scaling is almost certainly due to not simulating snow (in places erroneously). As a result I feel that the authors have over-stated their results that the model has strong-scaling and scales efficiently. I am also concerned that the model is not producing reasonable SWE.

Thank you referee #1 for your support and comments. We appreciate
your concern about running scaling experiments using 16 timesteps and
believe the new results displaying 2928 timesteps over a snow season
should suffice, especially given the more accurate depiction of CONUS
SWE.

L9 100's -> 100s (not possessive)

"1 m to 100s of meters"

L21 1800 cores contradicts 2304 listed above?

2304 was the maximum number of processes used for scaling, while 1800
was the number of processes used for the CONUS simulations based on
the Discover supercomputer architecture and resources. We will
clarify this difference.

L34 meters -> write out the order of magnitude. Just meters could be 1000s!

"(1-meter)"

L51 "can be" is a bit hedgy. I think this would be stronger to state what aspects of
snowmodel result in it being computationally expensive — physically based, 2 layer snow
model with energy balance with lateral transport.

"Physically derived snow algorithms, like SnowModel, that model the
energy balance, multilayer snow physics, and lateral snow transport
are computationally expensive."

L71 dimensional?

"…where each grid cell represents a point, or a one-dimensional
snowpack model, that is not influenced by nearby grid cells)"

L89 "properties" rather, these are states and fluxes

to model snow states (e.g., snow depth, SWE, snow melt, snow
density) and fluxes over different landscapes and climates

L91 This is unclear — is parallel input the only thing holding it back?

Thanks for your comment. We changed the wording to the following:

```
While many snow modeling systems exist, SnowModel will benefit from
parallelization because of its ability to simulate snow processes on a high
resolution grid through downscaling meteorological inputs and modeling snow
redistribution.
```

L104 missing closing ]

```
adding in missing ]
```

L131 The 23-24 period is unclear. It is perhaps made more clear in the results section, but
my notes here were asking if this was the sim period or just a subset of the full year
extracted? If the former, what are the initial conditions?

L147 "we hope to" I would be more firm in "we show" or similar

```
"we show its ability to efficiently run regional to continental sized
simulations."
```

L166 "CAF syntax…" not clear that this ads much — other aspects of Fortran syntax are not
noted. Is this just for algorithm readability later on? If the authors keep this, I suggest
tightening this section as much as possible

```
After reviewing both of the referee's comments, we ended up deleting
most of the algorithm references within the figures to simplify some
of the details. Therefore, I also deleted this section and figure
because it became less relevant. Thanks.
```

L195 Throughout, "process's" should be "process'"  as per -> "possessive of a plural noun is
formed by adding only an apostrophe when the noun ends in _s_"

```
Changed, thank you.
```

L199 I know that HX has been defined by here, but I'd forgotten what this was and I would
suggest considering writing it out again. Or just keep writing it out.

```
Halo-exchange is no longer abbreviated as HX in manuscript, thanks.
```

L200 "images" -> processes

```
Changed, thanks.
```

L202 "some CAF implementations" Which ones? Why not just not support them / avoid
them?

We have found limitations with the Cray compiler. We didn't really want to call that out b/c other compilers may have similar issues.

L215 is this spatially variable? if not, how do you select a representative value for something like CONUS domain?

It is not spatially variable. We used a value of 200.0 for all simulation experiments. Further testing of this parameter was beyond the scope of the study.

L221 I would suggest using monospace fonts instead of italics to refer to algorithm variables

Implemented this change, thank you.

L230 I would clearly note it's slow because of the comms overhead + mem transfer

Thank you. Changed to the following.

However, synchronization statements have an associated cost of decreasing the speed and efficiency of an algorithm due to communication overhead and memory transfer and therefore should be minimized…

L260 What happens if there is a wind direction discontinuity between the HX boundaries?

See sentence in previous paragraph. "To calculate the saltation flux, SnowModel iterates over **continuous** sections (jstart and jend) of the same wind direction,…." The continuous section is determined by subsequent grid cells with the same wind direction at a given timestep, allowing for wind direction discontinuities.

L285 Why maintain the serial portion if it makes the parallel code less optimal?

Thanks for your comment. In response we added this sentence description:

A future goal of this work is to be able to merge the serial and parallel versions of the code into one code base that can be easily maintained and utilized by different users with different computational resources. Therefore, we want to maintain both centralized and distributed approaches. However, for optimal parallel performance over larger simulation domains,…

L286 Reading past here I think I figured out centralized, but it's not super clear. My notes at this point were confused. The coorindation of all the processes working on this is not very clear to me and would benefit from a description.

Thanks for noting this confusion. Here is the text that describes the
difference between centralized and distributed. It also references
Figure 8.

Parallel implementations that are less memory restricted commonly use local to
global mapping strategies, or a **centralized** approach for file I/O (Fig. 8a).
However, this approach requires that each process stores global arrays for
input and output variables and creates a substantial bottleneck as the domain
size scales (Sect. 4.2). To improve performance, **distributed** file I/O can be
implemented, where input and output files are directly and concurrently
assessed by each process (Fig. 8b).

L297 I'm not sure describing the non-parallel ASCII files is worth while. Why not simply
state it needs binary files?

Changed paragraph to:

Parallel SnowModel's primary static spatial inputs include topography and
vegetation data. However, depending on the simulation configuration, additional
spatial inputs representing gridded values of latitude and longitude may be
required. While acceptable static input file types include binary, netcdf, and
ASCII files for the serial version of the code, optimizing the efficiency of
Parallel SnowModel requires static inputs from binary or netcdf files that can
be accessed concurrently by indexing the starting byte and length of bytes
commensurate to a process local domain. Therefore, each process only reads its
own portion of the static input data. The CONUS simulation could not be
simulated using a centralized approach because each process would be holding
global arrays of topography and vegetation in memory, each of which would be
approximately 5.2 GB of memory.

L300 process'

Changed, thank you.

L315 How slow/bottleneck/compute intensive is this step?

It depends on the spatial and temporal resolution of the simulations.

L333 this needs code coverage to convince the reader that the compute intensive code
paths have been stressed such that these are representative results.

Now that the Derecho supercomputer is accessible, we are able to
execute simulations faster and thus achieve longer simulations within
the 12-hour wall clock. Therefore, we anticipate being able to run
the CO Headwaters validation experiment for 5 months and thus provide
better validation confidence. We will run a CO Headwaters experiment
from February through June to account for both periods of

accumulation and ablation.

L364 I struggled with this section to understand what code version  was what, how it was
related to the final code, how different it was, etc. Suggest cutting or at a minimum tighten
significantly. I would also move the methodology descriptions into the methodology section.

Thank you. We will move the experiment description into the
methodology section and tighten up the labeling to make it clearer.

L386 what are these different code versions?

We appreciate you voicing your confusion here. Previous code versions
were used to investigate the scaling and profiling journey of the
code through its development. These versions are referenced in the
github. The "current" code is the latest version of the code within
the github. This version is used for the scaling analysis across the
different domains. We will make this more clear as we add more proper
methodology for the simulation experiments.

L390 same code coverage criticism here

Thank you. See above comment.

L431 "SWE-melt" suggest "ablation"

Thank you. Changed to ablation.

L432 Good to validate in the future, but as noted above the results as presented do not look
right for mid winter across the northern US and especially Canada

See comment above about incorrect CONUS visualization.

L465 I believe this is over-stated

We will reassess this comment after we get the updated scaling
results back for experiments conducted over one year.

L526 Why are these scripts not available? It should be included so-as to make the
experiments reproducible. Where can one obtain the input met forcing?

Our apologies, We will include all relevant scripts needed to
generate the experiments.

**Referee #2**

In general, the manuscript is well written with clear objectives, meticulous methods, and results. The study introduced a novel parallelization method to accelerate the SnowModel and apply it to simulations on a larger scale, which carries significant scientific significance. However, I am concerned that the scientific reproducibility and presentation quality of this manuscript should be improved before any publication with standards expected for GMD. Below, I will provide detailed comments on each section:

```
Thank you for your support and suggestions for improvement.
```

**Section 2**: While it briefly introduced SnowModel and the authors' motivation for parallelization, I suggest separating the introduction to SnowModel into its own section and incorporating schematic diagrams of the model's structure. These diagrams would assist readers in understanding the parallelization strategies discussed in Section 3.3, and the "Parallelization Motivation" could be a subsection within Section 2.2.

```
Thank you for your comment. We included a diagram that reflects the
important submodules of SnowModel. We were a bit confused about the
other suggestions. Currently, Section 2 provides an introduction to
SnowModel with a page of text describing the model. Then Section 2.1
provides motivation for its parallelization. Then the text moves onto
Section 3 (i.e. the Methods). Are you suggesting splitting up Section
2 into Section 2.1 (SnowModel) and Section 2.2 (Parallel Motivation)
because currently Section 2.2 does not exist. After reading over both
referee's comments regarding confusion about the structure of the
paper, we propose the following changes.
```

```
1.  Introduction
2.  Background
    2.1.  SnowModel
    2.2.  Coarray Fortran
    2.3.  Model Domains
    2.4.  Parallelization Motivation
3.  Methods
    3.1.  Parallel Implementation
        3.1.1.  Partitioning Algorithm
        3.1.2.  Non-trivial Parallelization
            3.1.2.1.  Wind and Solar Radiation Models
            3.1.2.2.  Snow Redistribution
        3.1.3.  File I/O
            3.1.3.1.  Parallel Inputs
            3.1.3.2.  Parallel Outputs
    3.2.  Simulation Experiments
        3.2.1.  Parallel SnowModel Validation
        3.2.2.  Parallel SnowModel Performance
```

……

**Section 3**: This section provides a wealth of code examples and diagrams that effectively elucidate the parallelization methods. The readers with some programming background can easily grasp the details of the parallelization techniques. However, the Section 3 delves excessively into minutiae, potentially causing readers to become lost in the details. Consider shortening this section, focusing on key aspects.

```
Thank you for your comment. Section 3 was significantly simplified by
deleting extra content relating to CAF syntax and algorithms used in
some of the non-trivial parallelization techniques in hopes to not
lose the reader, while still providing information relevant to its
parallelization. Additionally, as discussed above, we added a
methodology section for the simulation experiments in Section 3.
```

**Section 4**: The results presented in this section are somewhat confusing, raising concerns about the scientific quality and reproducibility of the study. Firstly, there is an overabundance of content related to model setup and evaluation metrics, which should not be presented as results. Furthermore, compared to Section 4.2, Sections 4.1 and 4.3 provide insufficient results, with a suspicion of excessive elaboration to magnify their importance.

```
Thank you for your comment. We moved the description of the
experiments and evaluation metrics to Section 3.2. We are not sure
what is meant by "insufficient results". As discussed in response to
referee #1's comments, we have expanded the timing of the validation
and performance experiments to make the results more meaningful.
Additionally, we significantly simplified the text within the results
sections in hopes of not providing excessive elaboration. If there is
anything we missed here, please let us know.
```

In **Section 4.1**, the description of the model setup occupies a disproportionate amount of space. The data provided to support validation conclusions are overly simplistic, such as "All variables across all processes produced RMSE values of $10^{-6}$" (Lines 341-342). I would like to see more detailed model comparisons, preferably presented in graphical form. Otherwise, consider merging this section with others.

Thank you for your comment. We changed the text in Lines 341-342 to the following.

Comparing the serial output of each of the seventeen selected variables (see Appendix B for a list of those variables) to those of each experiment conducted with a different number of processes produced RMSE values of $10^{-6}$

Additionally, we don't feel a graphical representation would be appropriate when output results are identical. We could provide an image of distributed SWE from a serial and parallel simulation on April 1st and then show the difference. However, if the difference is effectively zero everywhere, then it doesn't make for a very interesting visualization.

In **Section 4.2**, the authors present code profiling and speedup plots for three different stages, but I couldn't discern specific differences between "Distributed High Sync" and "Distributed Low Sync." I attempted to find an explanation in Section 3.4 but failed. Without a more detailed explanation, readers will struggle to understand the scientific significance of these results. For instance, it would be helpful to clarify what code optimizations improved process communication and reduced wait times.

Thank you for your comment. We moved the methodology of this section to be within Section 3.2. Additionally, we included a more succinct description of the different versions [i.e. "Distributed High Sync", etc]. We think that will make the methods and results pertinent to this section much clearer.

**Section 4.3** displays spatial results and time series of SWE, but it lacks information on how other snow properties performed. To convincingly demonstrate that Parallel SnowModel successfully simulates distributed snow over CONUS, it is essential to provide additional output results for different variables.

Thank you for your comment. SnowModel is primarily used to simulate SWE. We will explore graphics that also look into snow density, sublimation, and melt. However, SnowModel does contain many other hydrologic variables of interest.

**Section 6**: This section extensively references the work of others and highlights the relevance of this study to their work. However, I believe this content would be better placed within the Discussion section. The Conclusions section should provide a comprehensive summary of the study's work and results, offer conclusive remarks, and state the research's significance without excessive referencing.

Thank you. We switched content from the Conclusions and Discussion Sections in response to this comment.

In conclusion, the manuscript requires further improvement to meet the publication requirements of the journal, particularly regarding scientific quality and presentation quality. I therefore conclude with a **major revision** and hope that the revised manuscript will address the above-mentioned issues.

Thank you referee #2 for your comments. We are undergoing major revisions to the structure of the manuscript in an attempt to enhance the scientific and presentation quality.

---

## Author Response (AR2)

**First Revisions**

**Referee #1**

This is a review of "Parallel SnowModel (v1.0): a parallel implementation of a Distributed Snow-Evolution Modeling System (SnowModel)" where the authors describe imporovements made to Snowmodel by including a distributed parallel scheme. These improvements allow for running over large spatial extents and for longer temporal periods.

Broadly, I like what the authors have done to use CAF to bring parallelism to an existing code. Hydrology has long lacked HPC-aware models and the science has been poorer for it.

However, I struggle to follow aspects of this manuscript and I do not find the scaling results convincing. Finally, the lack of any validation against observations gives me pause, especially given the SWE results shown are almost certainly wrong for large portions of the domain in Figure 11. I will detail these concerns below.

```
We appreciate your support of this work and concern for some of the results.
In terms of scaling, we reran the scaling results for a period from 2017-09-
01 to 2018-09-01 (2928 timesteps vs. the previous 16 timesteps) for more
realistic timing comparisons.

In terms of the lack of validation, your comment made us dig into simulated
SWE values from Figure 11 near Calgary in Alberta and identify a binning
issue with how the data is being visualized. Therefore, most of Alberta
should be represented as the 0.01 – 0.10m SWE bin. Thank you very much for
this comment and apologies for the error. We updated the figure but
acknowledge that it still might be error prone. However, this study focuses
on parallelizing the algorithm, verifying that results are identical to
simulations using the serial version of the code and demonstrating
performance through strong scaling and CONUS simulations. We added additional
sentences referencing how extensively the serial code has been validated (see
approx. lines 331-341 in revised draft). However, it is beyond the scope of
this study to adequately validate Parallel SnowModel output on a CONUS scale.
```

First, the introduction essentially fails to cite any of the European or Canadian literature on snow dynamics, blowing snow, and the existing model developments that have been made. Notable contributions from Mott, Durand, Lehning, Vionnet, Marsh, Pomeroy, Musselman, MacDonald, Morin, Fang, and Essery to name but a few are all missing and would provide valuable context to the Liston, et al modelling efforts.

```
Thank you for addressing this oversight. We added the following paragraph to
the introduction (starting at approx. line 48 in revised draft).
```

Many physical snow models have been developed either in stand-alone algorithms or larger LSMs with varying degrees of complexity based on their application. The more advanced algorithms attempt to accurately model snow properties at higher resolution especially in regions where snow interacts with topography, vegetation, and/or wind. Wind-induced snow transport is one such complexity of snow that represents an important interaction between the cryosphere and atmosphere. It occurs in regions permanently or temporarily covered by snow and greatly influences snow heterogeneity, sublimation, avalanches, and melt timing. Models that have incorporated wind-induced physics generally require components to both develop the snow mass balance and incorporate atmospheric inputs of the wind field. However, there often exists a trade-off between the accuracy of simulating wind-induced snow transport and the computational requirements for downscaling and developing the wind fields over the gridded domain (Reynolds et al., 2021; Vionnet et al., 2014). Therefore, simplifying assumptions of uniform wind direction has been applied in models like Distributed Blowing Snow Model (DBSM) (Essery et al., 1999; Fang and Pomeroy, 2009). More advanced models have utilized advection-diffusion equations, like Alpine3D (Lehning et al., 2006) or spatial distributed formulations like SnowTran-3D (Liston and Sturm, 1998). Finite volume methods for more efficiently discretizing wind fields have been applied to models such as DBSM (Marsh et al., 2020). The most complex models consider nonsteady turbulence which utilize three-dimensional wind fields from atmospheric models to simulate blowing snow transport and sublimation; for example, SURFEX in Meso-NH/Crocus (Vionnet et al., 2014; Vionnet et al., 2017), wind fields from the atmospheric model ARPS (Xue et al., 2000) being incorporated into Alpine3D (Mott and Lehning, 2010; Mott et al., 2010; Lehning et al., 2008), and SnowDrift3D (Prokop and Schneiderbauer, 2011). Incorporating wind-induced physics into snow models is computationally expensive; thus, parallelizing the serial algorithms would likely be beneficial to many models.

Secondly, I find the mixing of methods and results to be very confusing. This is exceptionally bad in the Parallel Performance (S4.2) section where multiple code revisions are described. It is not at all clear where the different 'Distributed high Sync', etc are coming from. In some of these, the results presented are trivial — of course one would expect increased synchronization across more processes to incure scaling limitations. It is not clear if the SnowTran-3D plateau at 36-processes is the final code, or was a WIP code. I get the impression the authors are attempting to convey their profiling journey to optimize their code, but a) a general audience likely is not interested in all the specifics and b) it's confusing laid out leaving an interested reader muddled. For example on line 386, it is unclear /what/ versions of the code were even used. This section strikes me as the crux of the results, and is therefore an important section. However, I struggled to make my way through it. I would strongly sugges the authors split the methodology out and well describe what was profiled, etc and how this shaped the CAF implimentation. And then in the results, clearly and simply show "it is faster by XYZ for domains PQR".

Thank you for addressing this confusion, and we agree that the methods and results could be structured in a clearer way. We restructured the methods to more briefly discuss the parallelization approach (e.g. 3.1 Parallel Implementation) and then introduce the different simulation experiments before discussing Results in Section 4. An updated structure for the methods and results is as follows:

3. Methods
    3.1. Parallel Implementation
        3.1.1. Partitioning Algorithm
        3.1.2. Non-trivial Parallelization
            3.1.2.1.        Topography – Wind and Solar Radiation
            3.1.2.2.        Snow Redistribution
        3.1.3. File I/O
            3.1.3.1.        Parallel Inputs
            3.1.3.2.        Parallel Outputs
    3.2. Simulation Experiments
        3.2.1. Parallel Performance
            3.2.1.1.        Parallel Improvement
            3.2.1.2.        Strong Scaling
            3.2.1.3.        CONUS Simulations
4. Results
    4.1. Parallel Improvement
    4.2. Strong Scaling
    4.3.        CONUS Simulations

After restructuring the methods and results, we simplified the Code Progression and Parallel Improvement to hopefully reduce the confusion. However, we do believe that these are important tools and results that can help others in their journey to parallelize legacy code.

In addition, the 16 timesteps are really not compelling as currently presented. I am sympathetic to the computational constraints. However, without code coverage, is there any guarantee that the code was tested in a representative manner? For example, if there were few or no melt / blowing snow (or if there was any snow!) then the results would not be typical of a run. This criticism exists for the 1 month serial v. distributed period (L333) as well. Is this a representative period of time viz a viz excercising the toughest numerical code

paths (e.g., blowing snow and multilayer snowpacts, canopy interception) and highest sync code paths?

Thank you for this concern. We very much agree that this is not the best
representation of scaling results. Therefore, we ran the six domains for 2928
timesteps (2017/09/01 - 2018/09/01) instead of the previous 16 timesteps.
While this will affect the minimum number of processes that can be used, we
believe this will be representative of sufficient synchronization
opportunities across all domains.

My read is that Figure 10 is the "final" code that is evaluated for scaling testing. My following comments are through this lens. I do not find Figure 10 convincing of strong scaling. I would expect PNW to be the most difficult to simulate region with deep snow covers, and many blowing snow events. It performs weakly, with essentially plateaued scaling at 750 processes. As more non-blowing snow (and non-snow) cells are added in the CONUS domain, the scaling increases (shown in Figure 11). Essentially my read is the more non-snow cells that are added, the better the scaling. This is not a strong scaling result. Rephrased, over domains with significant snow processes, the scaling is poor.

Correct, Figure 10 (Figure 9 in the revised draft) represents the "final"
code. We will make that clearer as we restructure the methodology for the
simulation experiments. We reran the scaling results for 2928 timesteps
(2017/09/01 - 2018/09/01) instead of the previous 16 timesteps. These new
scaling results demonstrate that both the PNW and Western US domains scale
very similarly. The PNW domain has an approximate speedup of 555 running on
720 processes and plateaus closer to 2304 processes with a speedup of 941.
Therefore, we believe that the parallel code is scaling across state,
regional, and continental domains.

The simulated SWE results presented in Figure 11 are  suspect. This is total SWE on the ground in Feb, correct? In the middle of the winter (Feb) there is snow covering much of Canada — the foothills of Alberta, the Priaries of AB, SK, MB, and the Boreal forests of AB, SK, and MB. In the simulation results shown in Figure 11a, the domain east of continental divid, including the eastern Rockies, is shown as having zero SWE. This is almost certainly not correct. The authors note that an evaluation of the SWE data will be done at a later point, but if this number of no-op grid cells are being used for the scaling evaluation, then the scaling evaluation is not representative of a real winter simulation.

As mentioned, we really appreciate this insight as it pointed us to a bug in
our binning algorithm for visualizing the results. We updated the distributed
plots of SWE figures (a) and (b) to reflect the correct color visualization.
As a result, we are observing SWE values between 0.01 and 0.10 m outside of
Calgary and much of Alberta, for example. We also appreciate the implications
of not simulating SWE correctly as it comes to scaling due to potential
communication requirements across processes. However, validating Parallel

SnowModel for the simulated period across CONUS is outside the scope of this
study. We focused on validating the serial and parallel versions of the code
and highlighting the many other studies that have validated the serial
version code to observations (see approx. lines 331-341 in revised draft).
There can be many causes of a poor-quality simulation including forcing data,
model formulation, model input parameters, and more.  We are not evaluating
that in this study. Snow is present in most grid cells, and regardless of the
distribution of snow, the current study illustrates how well the current code
scales when running the domains selected.

Figure 11e shows the erosion and then deposition across a ridgeline. However, in most
mountain regions, this deposited snow will avalanche to a lower elevation. Given there is no
avalanche model in this code and no avalanche literature is cited, these results are not
compelling. Perhaps this is a ridgeline that doesn't have avalanches. But this needs to be
noted if true.

Thank you for that observation. Correct, SnowModel does not contain an
avalanche model. As a result, we added the following text in the revised
draft (approx. lines 484 – 486) "It is important to note that while SnowModel
does simulate snow redistribution, it does not currently have an avalanche
model, which may be a limitation of accurately simulation SWE within this
sub-domain".

In conclusion, I like that the authors are describing making the code HPC-aware by using
CAF with a simple halo exchange. I think there is value in showing the community that
"legacy" models can be updated and that it is "not that hard." Such messaging has the
potential to help normalize HPC-aware code development. However, the scaling results
seem to show significant limitations in the scaling and the better CONUS scaling is almost
certainly due to not simulating snow (in places erroneously). As a result I feel that the
authors have over-stated their results that the model has strong-scaling and scales
efficiently. I am also concerned that the model is not producing reasonable SWE.

Thank you referee #1 for your support and comments. We appreciate your
concern about running scaling experiments using 16 timesteps and believe the
new results displaying 2928 timesteps over a snow season should suffice,
especially given the more accurate depiction of CONUS SWE.

L9 100's -> 100s (not possessive)

We have made this change, thank you.

L21 1800 cores contradicts 2304 listed above?

2304 was the maximum number of processes used for in the original scaling
experiments (the revised scaling experiments used up to 3456 processes),

while 1800 was the number of processes used for the CONUS simulations from
2000-2021 executed on the Discover supercomputer. We decided on to run the
long-term simulations using 1800 processes based on the scaling experiments
and the Discover supercomputer architecture. However, to simplify the text we
will on report efficiency numbers for 1800 processes to keep it consistent
and less confusing (see approx. lines 17-19 in revised draft). Thank you.

L34 meters -> write out the order of magnitude. Just meters could be 1000s!

We made this change, thank you.

L51 "can be" is a bit hedgy. I think this would be stronger to state what aspects of
snowmodel result in it being computationally expensive — physically based, 2 layer snow
model with energy balance with lateral transport.

We have added the following sentence (revised draft: approx. lines 70-71):

"Physically derived snow simulation algorithms, as used in SnowModel, that
model the energy balance, multilayer snow physics, and lateral snow transport
are computationally expensive."

L71 dimensional?

We changed this wording to the following text (revised draft: lines 91-92):

"…where each grid cell represents a point, or a one-dimensional snowpack
model, that is not influenced by nearby grid cells)"

L89 "properties" rather, these are states and fluxes

We changed this wording to the following text (revised draft: approx. lines
110-111):

"..to model snow states (e.g., snow depth, SWE, snow melt, snow density) and
fluxes over different landscapes and climates.."

L91 This is unclear — is parallel input the only thing holding it back?

Thanks for your comment. We changed the wording to the following (revised
draft: approx. lines 112-113):

While many snow modeling systems exist, SnowModel will benefit from
parallelization because of its ability to simulate snow processes on a high
resolution grid through downscaling meteorological inputs and modeling snow
redistribution.

L104 missing closing ]

```
adding in missing ]
```

L131 The 23-24 period is unclear. It is perhaps made more clear in the results section, but my notes here were asking if this was the sim period or just a subset of the full year extracted? If the former, what are the initial conditions?

L147 "we hope to" I would be more firm in "we show" or similar

```
Thank you, we have changed the sentence to:
```

```
"we show its ability to efficiently run regional to continental sized
simulations."
```

L166 "CAF syntax…" not clear that this ads much — other aspects of Fortran syntax are not noted. Is this just for algorithm readability later on? If the authors keep this, I suggest tightening this section as much as possible

```
After reviewing both referee's comments, we ended up deleting most of the
algorithm references within the figures to simplify some of the details.
Therefore, I also deleted this section and figure because it became less
relevant. Thanks.
```

L195 Throughout, "process's" should be "process'" as per -> "possessive of a plural noun is formed by adding only an apostrophe when the noun ends in _s_"

```
Changed, thank you.
```

L199 I know that HX has been defined by here, but I'd forgotten what this was and I would suggest considering writing it out again. Or just keep writing it out.

```
Halo-exchange is no longer abbreviated as HX in manuscript, thanks.
```

L200 "images" -> processes

```
Changed, thanks.
```

L202 "some CAF implementations" Which ones? Why not just not support them / avoid them?

```
We have found limitations with the Cray compiler. We didn't really want to
call that out b/c other compilers may have similar issues, and ultimately the
Cray compiler can have these limits reduced with a lot of adjustments to
poorly documented environment variables and compiler flags, but it is a lot
to go into here. There are only 3 widely used CAF implementations that we are
aware of (Cray, Intel and Gnu), all have different tradeoffs. It feels like
these details are not in keeping with the theme of the paper.
```

L215 is this spatially variable? if not, how do you select a representative value for something like CONUS domain?

It is not spatially variable. We used a value of 200.0 for all simulation experiments. Further testing of this parameter was beyond the scope of the study.

We have added the following sentence (revised draft: approx. lines 231-232):

Future work should permit this parameter to vary spatially to account for changes in the length scale across the domain.

L221 I would suggest using monospace fonts instead of italics to refer to algorithm variables

Implemented this change, thank you.

L230 I would clearly note it's slow because of the comms overhead + mem transfer

Thank you. We ended up deleting this section to simplify the paper but added the following statement later in the draft when synchronization is introduced (see revised draft: approx. lines 351-353).

"Synchronization statements have an associated cost of decreasing the speed and efficiency of an algorithm due to communication overhead and memory transfer."

L260 What happens if there is a wind direction discontinuity between the HX boundaries?

See sentence in previous paragraph (approx. lines 249-252 in revised draft). "To calculate the final saltation flux (updated flux), SnowModel steps through regions of continuous wind direction (delineated by the indices: jstart and jend),…."

The continuous section is determined by subsequent grid cells with the same wind direction at a given timestep, allowing for wind direction discontinuities. Snow will accumulate where such convergent discontinuities exist regardless of the location of a halo.

L285 Why maintain the serial portion if it makes the parallel code less optimal?

Thanks for your comment. In response we added this sentence description (approx. lines 281-286 in revised draft):

A goal of this work was to combine the serial and parallel versions of the code into one code base that can be easily maintained and utilized by previous, current, and future SnowModel users with different computational resources and skills. Therefore, we want to maintain both the *Centralized* and

*Distributed* file I/O approaches. However, for optimal parallel performance over larger simulation domains, file input (reading) is performed in a *Distributed* way for the static inputs and in a *Centralized* way for dynamic inputs, while file output (writing) is performed in a *Distributed* way, as described further below.

L286 Reading past here I think I figured out centralized, but it's not super clear. My notes at this point were confused. The coorindation of all the processes working on this is not very clear to me and would benefit from a description.

Thanks for noting this confusion. We updated the text (and Figure 7 in the revised draft)to hopefully make this clearer. Below is the text updated text (approx. lines 269-274):

File I/O management can be a significant bottleneck in parallel applications. Parallel implementations that are less memory restricted commonly use local to global mapping strategies, or a **_Centralized_** approach for file I/O (**Error! Reference source not found.**a). This approach requires that one or more processes stores global arrays for input variables and that one process (Process 1; **Error! Reference source not found.**a) stores global arrays for all output variables. As the domain size increases, the mapping of local variables to global variables for outputting creates a substantial bottleneck. To improve performance, **_Distributed_** file I/O can be implemented, where input and output files are directly and concurrently accessed by each process (**Error! Reference source not found.**b).

L297 I'm not sure describing the non-parallel ASCII files is worth while. Why not simply state it needs binary files?

Changed paragraph to the following (approx. lines 292-299 in revised draft):

As noted above, SnowModel has two primary types of input files, temporally static files such as vegetation and topography and transient inputs such as meteorological forcing data. While acceptable static input file types include flat binary, NetCDF, and ASCII files for the serial version of the code, optimizing the efficiency of Parallel SnowModel requires static inputs from binary files that can be accessed concurrently and directly subset by indexing the starting byte and length of bytes commensurate to a process local domain. Therefore, each process can read its own portion of the static input data. For very large domains, the available memory becomes a limitation when using the centralized approach. For example, the CONUS simulation could not be simulated using a centralized file I/O approach because each process would be holding global arrays of topography and vegetation in memory, each of which would require approximately 5.2 GB of memory per process.

L300 process'

```
Changed, thank you.
```

L315 How slow/bottleneck/compute intensive is this step?

```
It depends on the spatial and temporal resolution of the simulations. For
example, running the post processing script on one output variable for the CO
Headwaters scaling simulations over a water year took approximately 6 minutes
to concatenate on a single core. We decided not to add a reference to this to
not further complicate this section.
```

L333 this needs code coverage to convince the reader that the compute intensive code paths have been stressed such that these are representative results.

```
Based on other comments we received, we ended up removing the Parallel
SnowModel Validation section to Appendix B. We were not able to rerun
validation experiments over the CO Headwaters domain for a longer period
based on computational limitations. It was impossible to run a serial
simulation over large domains for long periods within the computer's 12-hour
wallclock.
```

L364 I struggled with this section to understand what code version was what, how it was related to the final code, how different it was, etc. Suggest cutting or at a minimum tighten significantly. I would also move the methodology descriptions into the methodology section.

```
Thank you. We moved the experiment description into the methodology section
and tightened up the labeling to make it clearer.
```

L386 what are these different code versions?

```
We appreciate you voicing your confusion here. Previous code versions were
used to investigate the scaling and highlight bottlenecks in the code through
its parallelization. These versions are referenced in the GitHub repository.
We changed the labeling and description of these different versions and
believe this section is now easier to follow.
```

L390 same code coverage criticism here

```
Thank you. See above comment.
```

L431 "SWE-melt" suggest "ablation"

Ablation refers to the combination of melt, sublimation, and other processes that remove snow.  In this case we are specifically referring to melt.

L432 Good to validate in the future, but as noted above the results as presented do not look right for mid winter across the northern US and especially Canada

See comment above about incorrect visualization.

L465 I believe this is over-stated

Based on our updated scaling results, we believe this statement is appropriate.

L526 Why are these scripts not available? It should be included so-as to make the experiments reproducible. Where can one obtain the input met forcing?

Our apologies, we included all relevant scripts needed to generate the experiments.

**Referee #2**

In general, the manuscript is well written with clear objectives, meticulous methods, and results. The study introduced a novel parallelization method to accelerate the SnowModel and apply it to simulations on a larger scale, which carries significant scientific significance. However, I am concerned that the scientific reproducibility and presentation quality of this manuscript should be improved before any publication with standards expected for GMD. Below, I will provide detailed comments on each section:

Thank you for your support and suggestions for improvement.

**Section 2**: While it briefly introduced SnowModel and the authors' motivation for parallelization, I suggest separating the introduction to SnowModel into its own section and incorporating schematic diagrams of the model's structure. These diagrams would assist readers in understanding the parallelization strategies discussed in Section 3.3, and the "Parallelization Motivation" could be a subsection within Section 2.2.

Thank you for your comment. We included a diagram that reflects the important submodules of SnowModel. We were a bit confused about the other suggestions. Currently, Section 2 introduces SnowModel with a page of text describing the model. Then Section 2.1 provides motivation for its parallelization. Then the text moves onto Section 3 (i.e. the Methods). Are you suggesting splitting up Section 2 into Section 2.1 (SnowModel) and Section 2.2 (Parallel Motivation) because currently Section 2.2 does not exist. After reading over both referee's comments regarding confusion about the structure of the paper, we made the following changes.

 1.   Introduction

……

**Section 3**: This section provides a wealth of code examples and diagrams that effectively elucidate the parallelization methods. The readers with some programming background can easily grasp the details of the parallelization techniques. However, the Section 3 delves excessively into minutiae, potentially causing readers to become lost in the details. Consider shortening this section, focusing on key aspects.

Thank you for your comment. Section 3 was significantly simplified by deleting extra content relating to CAF syntax and algorithms used in some of the non-trivial parallelization techniques in hopes to not lose the reader, while still providing information relevant to its parallelization. Additionally, as discussed above, we added a methodology section for the simulation experiments in Section 3.

**Section 4**: The results presented in this section are somewhat confusing, raising concerns about the scientific quality and reproducibility of the study. Firstly, there is an overabundance of content related to model setup and evaluation metrics, which should not be presented as results. Furthermore, compared to Section 4.2, Sections 4.1 and 4.3

provide insufficient results, with a suspicion of excessive elaboration to magnify their importance.

Thank you for your comment. We moved the description of the experiments and evaluation metrics to Section 3.2. We are not sure what is meant by "insufficient results". As discussed in response to referee #1's comments, we have expanded the timing of the scaling experiments to make the results more meaningful. Additionally, we significantly simplified the text within the results sections in hopes of not providing excessive elaboration. If there is anything we missed here, please let us know.

In **Section 4.1**, the description of the model setup occupies a disproportionate amount of space. The data provided to support validation conclusions are overly simplistic, such as "All variables across all processes produced RMSE values of $10^{-6}$" (Lines 341-342). I would like to see more detailed model comparisons, preferably presented in graphical form. Otherwise, consider merging this section with others.

Thank you for your comment. We merged the text into Section 3.2 Simulation Experiments and it reads (approx. lines 325-330):

Validation experiments comparing output from the original serial version of the code to the parallel version were conducted continuously during the parallel algorithm development to assess the reproducibility of the results. Additionally, an extensive validation effort was performed at the end of the study that compared output of the serial algorithm to that of the parallel algorithm, while varying the domain size, the number of processes, and therefore the domain decomposition. Results from these validation experiments produced root mean squared error (RMSE) values of $10^{-6}$, which is at the limit of machine precision, when compared to serial simulation results. See Appendix B for more details on the validation experiments.

Additionally, we don't feel a graphical representation would be appropriate when output results are identical to within the expectations of numerical precision. We could provide an image of distributed SWE from a serial and parallel simulation on April 1st and then show the difference. However, if the difference is effectively zero everywhere, then it does not make for a very interesting visualization.

In **Section 4.2**, the authors present code profiling and speedup plots for three different stages, but I couldn't discern specific differences between "Distributed High Sync" and "Distributed Low Sync." I attempted to find an explanation in Section 3.4 but failed. Without a more detailed explanation, readers will struggle to understand the scientific significance of these results. For instance, it would be helpful to clarify what code optimizations improved process communication and reduced wait times.

Thank you for your comment. We moved the methodology of this
section to be within Section 3.2. Additionally, we included a more
succinct description of the different versions. We think that will
make the methods and results pertinent to this section much
clearer. The different versions reference the centralized and
distributed file I/O schemes discussed earlier in Section 3.1.3 and
the final code version at the time of this publication. These
different versions can also be referenced in the GitHub repository.

**Section 4.3** displays spatial results and time series of SWE, but it lacks information on how
other snow properties performed. To convincingly demonstrate that Parallel SnowModel
successfully simulates distributed snow over CONUS, it is essential to provide additional
output results for different variables.

Thank you for your comment. SnowModel is primarily used to simulate
SWE. Our validation experiments compared output of sixteen snow
variables using the serial and parallel versions of the code. A
list of those variables can be found in Appendix B. This Section is
focused on demonstrating that the parallelization effort can
produce distributed snow variables at the CONUS scale. Future work
will involve validating this dataset, but our validation focus was
making sure that the serial and parallel versions produced the same
results.

**Section 6**: This section extensively references the work of others and highlights the
relevance of this study to their work. However, I believe this content would be better placed
within the Discussion section. The Conclusions section should provide a comprehensive
summary of the study's work and results, offer conclusive remarks, and state the research's
significance without excessive referencing.

Thank you. We switched content from the Conclusions and Discussion
Sections in response to this comment.

In conclusion, the manuscript requires further improvement to meet the publication
requirements of the journal, particularly regarding scientific quality and presentation quality.
I therefore conclude with a **major revision** and hope that the revised manuscript will
address the above-mentioned issues.

Thank you referee #2 for your comments. We are undergoing major
revisions to the structure of the manuscript to enhance the
scientific and presentation quality.

**Second Revisions**

**Referee #1**

None
**Referee #2**

None
**Referee #3**

Review of "Parallel SnowModel (v1.0): a parallel implementation of a Distributed Snow-Evolution Modeling System (SnowModel)" by Mower et al.

Preliminary note to the editor and authors: My expertise includes snowpack modelling, but without particular expertise in computational methods such as parallelization. Please consider my assessment of the manuscript cannot be complete.

The authors present a parallel implementation of the widely used SnowModel, a description of used methods, and an assessment of scaling performance, using snowpack simulations for one year over domains of different sizes, up to the contiguous United States. This development effort is important for the snow modelling community as it enables to run seasonal (or multi-seasonal) snowpack simulations over large domains (e.g., necessary for several operational applications), while resolving small-scale processes such as wind-induced snow transport. The revised manuscript is clear and convincing (within the limits of my expertise, as specified above). I believe the authors have addressed most of the concerns of the previous reviewers. Please find below a few technical remarks.

`We appreciate your support. See corrections to the technical remarks below.`

l. 49: high resolution (or higher than?).

`This was a typo, we changed it to "high".`
In this paragraph, mention the order of magnitude of spatial resolution required to model wind-induced snow transport, which justifies your use of a 100 m resolution in this study.

`Added the following sentence in the updated version of the manuscript (approx. lines 56-57), "Additionally, these models typically require high resolution grids (1 to 100m) as the redistribution components of the model become negligible at larger spatial discretizations (Liston et al., 2007).`
l. 52-53: it could be worth citing at least one reference paper about the influence of wind-induced snow transport on the snowpack distribution.

We agree and thanks for pointing this out. We changed the following
sentence (approx. lines 50-54 in updated draft), "Wind-induced snow
transport is one such complexity of snow that represents an
important interaction between the cryosphere and atmosphere. It
occurs in regions permanently or temporarily covered by snow,
influences snow properties (e.g. heterogeneity, sublimation,
avalanches, and melt timing), and has been shown to improve
simulated snowpack distribution (Bernhardt et al., 2012; Freudiger
et al., 2017; Keenan et al., 2023; Quéno et al., 2023).

Sharma et al. (2004) reference is incomplete in the reference section.

Thanks for noting this. We were unable to find a more complete
reference on google scholar or elsewhere on the internet and thus
removed the citation since we were already citing other studies.

Figure 1: typo in the name Pedersen. Please also reformulate the caption: the original figure
from Pedersen et al. (2015) was modified for the present paper, and not "modified by
Pederson et al. (2015)" (if I understand correctly).

Thanks for identifying this. We implemented your recommended
correction.

Figure 11: consider changing the unit of SWE to mm which is the common scientific unit
used for this variable, as it is equivalent to kg/m2.

Since "m" is a standard SI unit and fits better with the plot
layout (changing to mm affects the y-axis tick labels and the
readability of the plot), we have decided to not implement this
change. However, we appreciate your comment.

Thank you referee #3 for your time and comments.